# Impact of a human gut microbe on *Vibrio cholerae* host colonization through biofilm enhancement

Kelsey Barrasso[1,2], Denise Chac[3], Meti D Debela[4], Catherine Geigel[5], Anjali Steenhaut[1], Abigail Rivera Seda[1], Chelsea N Dunmire[3], Jason B Harris[4,6], Regina C Larocque[4], Firas S Midani[7], Firdausi Qadri[8], Jing Yan[5,9], Ana A Weil[3]*, Wai-Leung Ng[1,2]*

[1]Department of Molecular Biology and Microbiology, Tufts University School of Medicine, Boston, United States; [2]Program of Molecular Microbiology, Graduate School of Biomedical Sciences, Tufts University School of Medicine, Boston, United States; [3]Department of Medicine, University of Washington, Seattle, United States; [4]Division of Infectious Diseases, Massachusetts General Hospital, Boston, United States; [5]Department of Molecular, Cellular and Developmental Biology, Yale University, New Haven, United States; [6]Department of Pediatrics, Harvard Medical School, Boston, United States; [7]Department of Molecular Virology and Microbiology at Baylor College of Medicine, Houston, United States; [8]International Center for Diarrheal Disease Research, Bangladesh, Dhaka, Bangladesh; [9]Quantitative Biology Institute, Yale University, New Haven, United States

*For correspondence:
anaweil@uw.edu (AAW);
wai-leung.ng@tufts.edu (W-LeungN)

Competing interest: The authors declare that no competing interests exist.

**Abstract** Recent studies indicate that the human intestinal microbiota could impact the outcome of infection by *Vibrio cholerae*, the etiological agent of the diarrheal disease cholera. A commensal bacterium, *Paracoccus aminovorans*, was previously identified in high abundance in stool collected from individuals infected with *V. cholerae* when compared to stool from uninfected persons. However, if and how *P. aminovorans* interacts with *V. cholerae* has not been experimentally determined; moreover, whether any association between this bacterium alters the behaviors of *V. cholerae* to affect the disease outcome is unclear. Here, we show that *P. aminovorans* and *V. cholerae* together form dual-species biofilm structure at the air–liquid interface, with previously uncharacterized novel features. Importantly, the presence of *P. aminovorans* within the murine small intestine enhances *V. cholerae* colonization in the same niche that is dependent on the *Vibrio* exopolysaccharide and other major components of mature *V. cholerae* biofilm. These studies illustrate that multispecies biofilm formation is a plausible mechanism used by a gut microbe to increase the virulence of the pathogen, and this interaction may alter outcomes in enteric infections.

## Editor's evaluation

In this work, the authors study the previously reported positive association between the presence of the gut bacterium *Paracoccus aminovorans* and *Vibrio cholerae* during infection. They describe and image dual-species biofilm formed in vitro as well as enhanced *V. cholerae* gut colonization in the presence of *P. aminovorans* in a neonatal mouse model. Collectively, the authors conclude that *P. aminovorans* enhances biofilm formation by *V. cholerae*, which could explain the increased abundance of *P. aminovorans* in persons infected with *V. cholerae* in endemic areas, and may have impacted the course of infections in humans.

## Introduction

*Vibrio cholerae* (*Vc*) causes an estimated 3 million infections and 120,000 deaths each year, and larger and more deadly outbreaks have increased during the last decade (*Camacho et al., 2018*; *Luquero et al., 2016*). A wide range of clinical outcomes occur in persons exposed to *Vc*, ranging from asymptomatic infection to severe secretory diarrhea. It is nearly certain that many behaviors of *Vc* in the aquatic environment and inside the host are significantly affected by the presence of other microbes (*Weil and Ryan, 2018*), and recent studies provide evidence that the gut microbiota may impact the severity of cholera (*Alavi et al., 2020*; *Hsiao et al., 2014*; *Levade et al., 2020*; *Midani et al., 2018*).

Several functions of the gut microbiota influence the growth or colonization of enteric pathogens, including production of antimicrobial compounds, maintenance of the intestinal barrier, regulation of the host immune response, and modulation of available nutrients (*McKenney and Pamer, 2015*). Gut microbes have been shown to have an important role in *Vc* infection in various animal models. For instance, disruption of the commensal microbiota with antibiotics is required to allow successful *Vc* colonization in adult rodent models. Starvation and streptomycin treatment are needed to reduce the intestinal normal flora to allow *Vc* infection to occur in the cecum and the entire bowel in guinea pigs (*Freter, 1955*). In another case, streptomycin treatment is needed to allow *Vc* colonization, mainly in the cecum and large intestine, of adult mice (*Nygren et al., 2009*). Conversely, *Vc* actively employs a type VI secretion system to attack host commensal microbiota to enhance colonization of the gut in infant mice (*Zhao et al., 2018*). Moreover, specific microbial species have a profound impact on *Vc* colonization. Pre-colonization with *Blautia obeum*, an anaerobic Gram-positive bacterium, in adult gnotobiotic mice decreases the *Vc* counts in the feces after infection. *B. obeum* is thought to produce a signaling molecule that induces *Vc* into a high cell-density quorum sensing state (*Hsiao et al., 2014*) in which virulence gene expression is repressed (*Jung et al., 2015*; *Watve et al., 2020*). Certain microbiota species reduce *Vc* colonization in germfree adult and suckling mice by producing the enzyme bile salt hydrolase that degrades the host-produced virulence-activating compound taurocholate (*Alavi et al., 2020*; *Hsiao et al., 2014*). Through metabolizing host glycans into short-chain fatty acids that suppress *Vc* growth, a prominent commensal species, *Bacteroides vulgatus,* reduces *Vc* proliferation within the intestine of germfree adult mice and typical laboratory infant mice (*You et al., 2019*).

While the above studies exemplify how a single microbe or a group of microbes can protect the host from *Vc* infection, the mechanisms used by certain gut microbes to promote *Vc* virulence, thereby increasing the likelihood of individuals developing cholera and worsening disease outcomes, are less well understood. We have previously studied household contacts of cholera patients to understand how gut microbes impact on susceptibility to cholera, and we identified bacteria associated with increased or decreased susceptibility to *Vc* infection (*Levade et al., 2020*; *Midani et al., 2018*). We also observed that the gut microbial species *Paracoccus aminovorans* (*Pa*) was more abundant in the gut microbiota during *Vc* infection (*Midani et al., 2018*). This association between *Pa* and *Vc* is unusual because most of the native gut microbiota is typically displaced by secretory diarrhea during cholera (*Hsiao et al., 2014*; *David et al., 2015*). To determine the underlying mechanisms driving these correlative clinical findings, we evaluated the relationship between *Pa* and *Vc* in co-culture and determined the effects of *Pa* on *Vc* infection outcomes with in vivo models. Here, we show that *Pa* interacts directly with *Vc* to form dual-species biofilm structures with previously uncharacterized features. Moreover, *Vc* colonization inside the animal host is enhanced by the presence of *Pa* in the small intestine, and this effect is dependent upon *Vc* biofilm production. Our findings demonstrate a plausible mechanism by which a gut microbe specifically associates with *Vc,* and this reinforces our microbiome analysis in humans that identified *Pa* as highly associated with infected individuals. Our findings also demonstrate that interactions between these two species have the potential to directly impact *Vc* pathogenesis and alter outcomes in human *Vc* infection.

## Results

### *P. aminovorans* is differentially abundant in individuals with active *V. cholerae* infection

*Paracoccus* is a genus of soil microbes found in low abundance in the gut microbiome of humans (*Yatsunenko et al., 2012*; *Urakami et al., 1990*). Our previously published analysis of stool gut

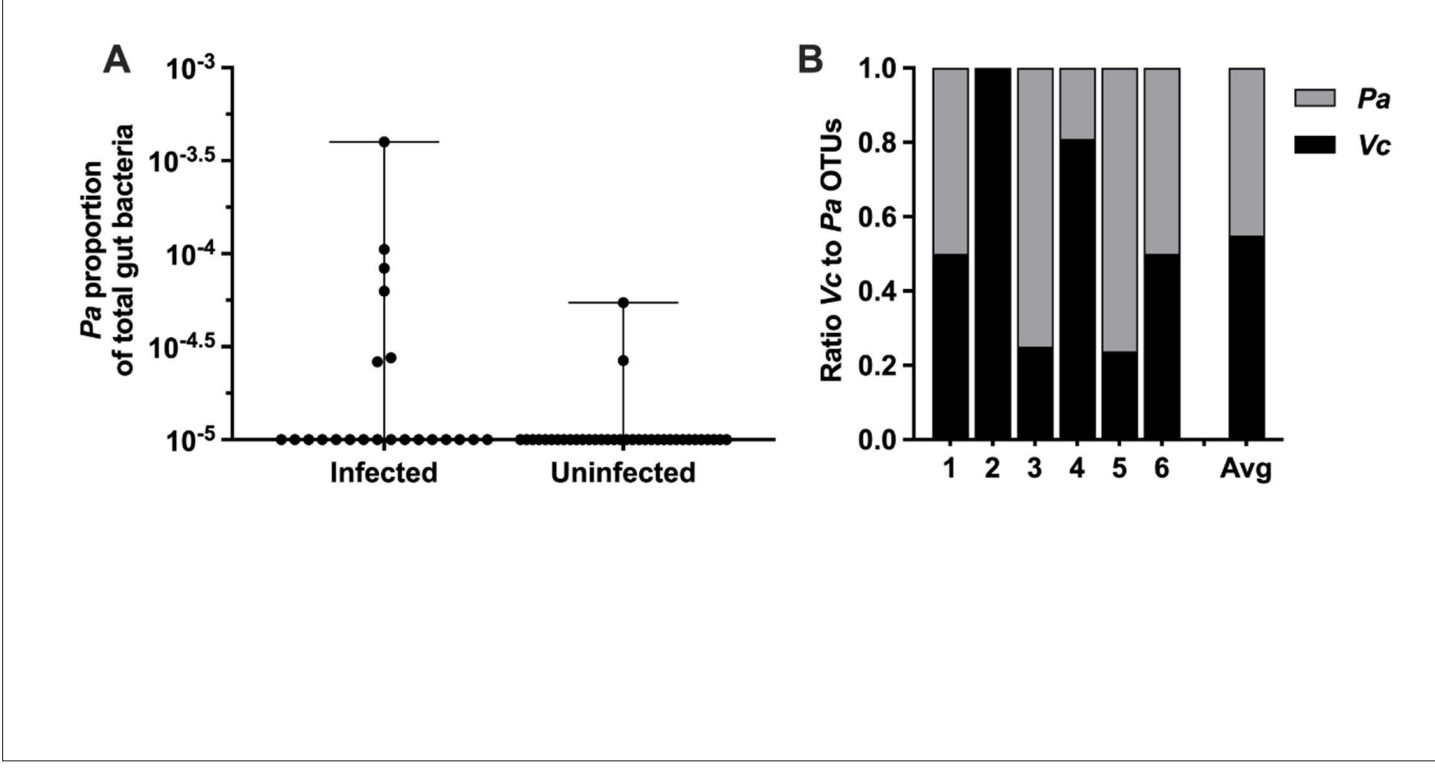

**Figure 1.** *Paracoccus aminovorans* (*Pa*) is more abundant in persons with *Vibrio cholerae* (*Vc*) infection compared to uninfected persons. In a prior study of household contacts of cholera patients in Bangladesh (***Midani et al., 2018***), *Pa* was identified as differentially abundant using a support vector machine model with recursive feature elimination in order to discriminate patterns of microbial taxa relative abundance that distinguished infected from uninfected persons. The microbiota was assessed using 16S rRNA in rectal swabs collected from individuals with *Vc* infection (n = 22) compared to uninfected individuals (n = 36). In this study, total sum normalization was applied to operational taxonomic unit (OTU) counts from each sample, and a median of 37,958 mapped reads per sample was generated (***Midani et al., 2018***). Based on this sequencing data, the estimated limit of detection for a *Pa* OTU is $2.0 \times 10^{-5}$. Raw and normalized counts of *Pa* and *Vc* are shown in ***Supplementary file 1***. (**A**) Normalized relative abundance of *Pa* in infected and uninfected individuals, comparison between infected and uninfected p=0.009 (Mann–Whitney nonparametric *U*-test). All data points are shown, and bars mark the maximum and minimum values. (**B**) Ratio of *Vc* to *Pa* in six *Vc*-infected persons.

The online version of this article includes the following source data for figure 1:

**Source data 1.** Paracoccus aminovorans (Pa) abundance in persons with and without *Vibrio cholerae* (Vc) infection.

**Source data 2.** Ratio of Paracoccus aminovorans (Pa) to *Vibrio cholerae* (Vc) in infected persons.

microbes from household contacts of cholera patients identified *Pa* as an unexpectedly abundant gut microbe during active *Vc* infection, and this organism was rarely found in uninfected participants (***Midani et al., 2018***). In this prior study, we used a support vector machine (SVM) model with recursive feature elimination to learn patterns of relative abundance of operational taxonomic units (OTUs) that distinguished infected (defined as *Vc* DNA detected in stool or a stool culture with *Vc* growth) from uninfected persons (*Vc* DNA undetected in stool and *Vc* stool culture negative). The model was trained on a subset of study participants and tested on another subset in a hold-out validation. Here, we have extracted data from this prior study to examine separately the *Pa* OTUs in infected compared to uninfected persons. *Pa* abundance was significantly higher as a proportion of the total sequencing reads in the stool of infected household contacts (6/22, 27%) compared to only 5.6% (2/36) of uninfected individuals (***Figure 1A***, p=0.009 by Mann–Whitney *U*-test). The ratio of *Pa* to *Vc* abundance present during infection was variable and averaged 1:1 (***Figure 1B***). These findings were particularly interesting because typically there is a drastic reduction of nearly all gut microbes during active *Vc* infection (***Hsiao et al., 2014***; ***David et al., 2015***) due to secretory diarrhea, oral rehydration solution ingestion, and *Vc* infection itself, and yet here *Pa* was found in an increased abundance in some actively infected participants. The *Vc* count in *Pa*-colonized infected household contacts compared to non-*Pa* colonized household contacts was modestly higher (approximately threefold) in this study;

however, statistically the difference was not significant (Mann–Whitney *U*-test, p=0.1474), which is likely due to small sample size (*Supplementary file 1*). Based on these findings, we hypothesized that *Pa* may be resistant to displacement from the gut during infection. While our previous study demonstrates a positive correlation between *Pa* in human stool and *Vc* infection, a causal relationship between this gut microbiota species and *Vc* infection had not been previously established.

## *Pa* increases *Vc* host colonization

We modified a well-established infant mouse colonization model (*Klose, 2000*) to assess whether the presence of *Pa* in the small intestine would promote *Vc* host colonization. First, we isolated a spontaneous streptomycin-resistant (Strep^R) mutant derived from the ATCC-type strain of *Pa* for selection and enumeration of *Pa* following host colonization. Infant mice (3-day-old) were intragastrically inoculated with *Pa* ($10^7$ colony-forming units [CFUs]) every 12 hr for four doses (0, 12, 24, and 36 hr). At 24 hr (i.e., 12 hr after the second *Pa* inoculation and right before the third *Pa* inoculation) and 48 hr (i.e., 12 hr after the last *Pa* inoculation), small intestines from these animals were dissected and homogenized. Gut homogenates were serially diluted and plated on medium containing streptomycin to assess *Pa* colonization. Strep^R *Pa* colonies ($>10^6$ CFUs/small intestine) were recovered at these two time points (*Figure 2A*), and no Strep^R colonies were detected in the mock-treated group, indicating that *Pa* successfully and stably colonized the small intestines of these animals using these methods at least for 12 hr. Unlike previous studies (*Freter, 1955*; *Nygren et al., 2009*), pretreatment with antibiotics was not required for *Pa* colonization (*Figure 2A*). Sequencing analysis of the mouse small intestines demonstrated no significant change in the microbial composition and diversity with and without *Pa* colonization (*Figure 2—figure supplement 1*).

We then evaluated if pre-colonization by *Pa* would influence *Vc* colonization in the small intestine. Four doses of *Pa* were inoculated into infant mice as described above. Negative control animals were inoculated with sterile media in place of *Pa* over the same dosing schedule. 12 hr after the last *Pa* inoculation (i.e., 48 hr after the first *Pa* inoculation), these animals were infected with *Vc* ($10^6$ CFU) to evaluate whether pre-colonization with *Pa* had an impact on *Vc* colonization. Although we do not fully understand the exact composition and growth dynamics of *Vc* and *Pa* inside the human gut, the pre-colonization/infection scheme was aimed to closely simulate the ratio of *Pa* to *Vc* observed in the gut microbiota of *Vc*-infected humans (*Figures 1 and 2A*). Comparing *Pa* pre-colonized mice to the control group, there was a significant increase (~10-fold, p≤0.0001) of *Vc* colonization in the mice pre-colonized with *Pa* (*Figure 2B*) 24 hr after infection. This enhanced intestinal colonization by *Vc* in the *Pa*-colonized mice was observed as early as 6 hr after infection and maintained throughout the colonization period (*Figure 2—figure supplement 2*). We did not monitor *Vc* counts in the small intestine beyond 24 hr after infection due to institutional restrictions. Using this infection model, similar to human infection, the *Vc:Pa* ratio inside the mouse intestine at the early stage of infection (6 hr after *Vc* infection, 18 hr after the last *Pa* inoculation) was variable, but was approximately 10:1 (*Figure 2—figure supplement 2*). We noticed that once *Pa* inoculation ended *Pa* abundance in the small intestine decreased over time but was still detectable 22 hr after the last inoculation (i.e., 10 hr after *Vc* infection; *Figure 2—figure supplement 2*). Thus, even though this animal model does not fully mimic *Vc* infection of human host, it allows a sufficient time to study the interaction between *Vc* and *Pa* in the intestinal environment.

We reasoned that it was also possible for *Vc* and *Pa* to encounter one another in the environment before entering the host. To model this scenario, *Vc* was mixed with *Pa* in a 1:1 ratio, and the mixture was used immediately for animal infection. In agreement with the results obtained with the *Pa* pre-colonization model, *Vc* intestinal colonization was significantly higher when coinfected with *Pa* than without *Pa* (*Figure 2C*). Given that *Pa* colonization did not overtly change the overall composition of the gut microbiota (*Figure 2—figure supplement 1*), collectively, our results demonstrate that the presence of a single gut microbiota species is sufficient to increase *Vc* host colonization. Our findings also illustrate that our approach to microbiome studies in humans (*Levade et al., 2020*; *Midani et al., 2018*) can be used as a predictive tool to identify gut microbes that alter *Vc* colonization.

## *Pa* promotes *Vc* biofilm formation

To investigate if the increased *Vc* intestinal colonization is due to direct interactions between *Vc* and *Pa*, these two species were co-cultured and allowed to propagate for 3 days where both planktonic

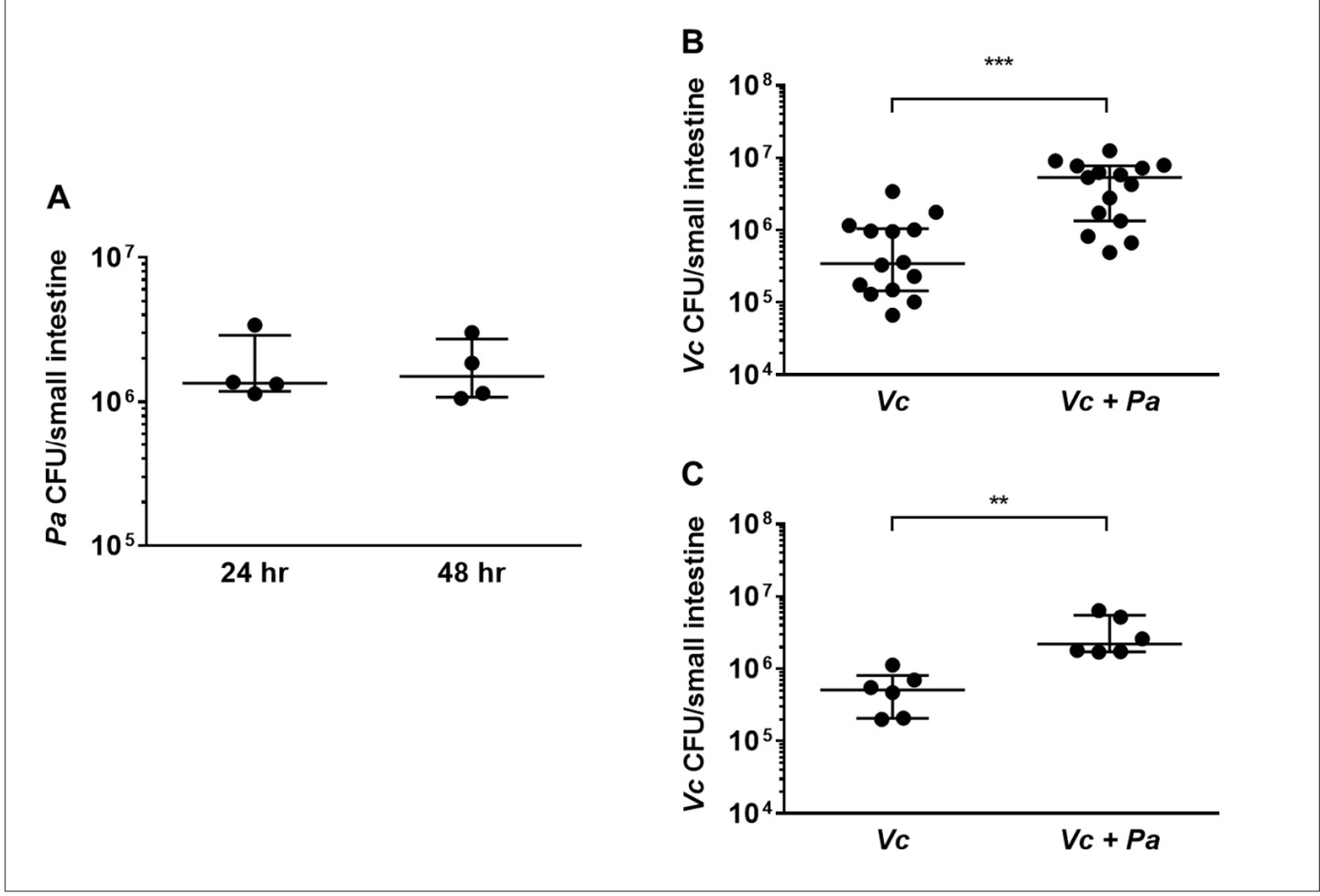

**Figure 2.** The presence of *Paracoccus aminovorans* (*Pa*) enhances *Vibrio cholerae* (*Vc*) colonization in the infant mouse intestine. (**A**) 3-day-old infant mice were intragastrically inoculated with $10^7$ colony-forming unit (CFU) of *Pa* every 12 hr twice or four times. 12 hr after the second or fourth dose of *Pa* inoculation (i.e., at 24 hr and 48 hr, respectively), mice were sacrificed and CFUs were enumerated by plating serial dilutions of small intestine samples on selective media. (**B**) 3-day-old infant mice were intragastrically inoculated four times with LB or $10^7$ CFU of *Pa* for every 12 hr, and 12 hr after the last *Pa* inoculation, the animals were infected with $10^6$ CFU of WT *Vc*. Mice were sacrificed 20–24 hr post infection, and the small intestine samples were taken to enumerate *Vc*. Bars on graphs depict median value with 95% confidence interval (CI) and individual data points plotted. Unpaired nonparametric *U*-test (Mann–Whitney); ***$p \le 0.001$. (**C**) *Vc* was inoculated intragastrically into the animals alone or together with *Pa* in a 1:1 ratio. After 24 hr, enumeration of *Vc* was performed as described above. Unpaired nonparametric *U*-test (Mann–Whitney); **$p \le 0.01$.

The online version of this article includes the following source data and figure supplement(s) for figure 2:

**Source data 1.** Viable cell count of Paracoccus aminovorans (Pa) in infant mouse intestine.

**Source data 2.** Viable cell count of *Vibrio cholerae* (Vc) in the infant mouse intestine with and without Pa pre-colonization.

**Source data 3.** Viable cell count of *Vibrio cholerae* (Vc) in the infant mouse intestine with and without Pa co-infection.

**Figure supplement 1.** *Paracoccus aminovorans* (*Pa*) colonization does not significantly alter the mouse gut microbial diversity.

**Figure supplement 1—source data 1.** Abundance of microbes in the mouse intestine with and without Pa colonization.

**Figure supplement 1—source data 2.** Principal Component Analysis of microbial abundance in mouse intestine with and without Pa colonization.

**Figure supplement 2.** *Paracoccus aminovorans* (*Pa*) colonization significantly increases *Vibrio cholerae* (*Vc*) small intestine colonization.

**Figure supplement 2—source data 1.** Viable cell count of *Vibrio cholerae* (Vc) in the mouse small intestine with and without Pa pre-colonization.

**Figure supplement 2—source data 2.** Viable cell count of Paracoccus aminovorans (Pa) and *Vibrio cholerae* (Vc) in the mouse small intestine 6 hours after Vc infection.

**Figure supplement 2—source data 3.** Viable cell count of Paracoccus aminovorans (Pa) in the mouse intestine with and without *Vibrio cholerae* (Vc) infection.

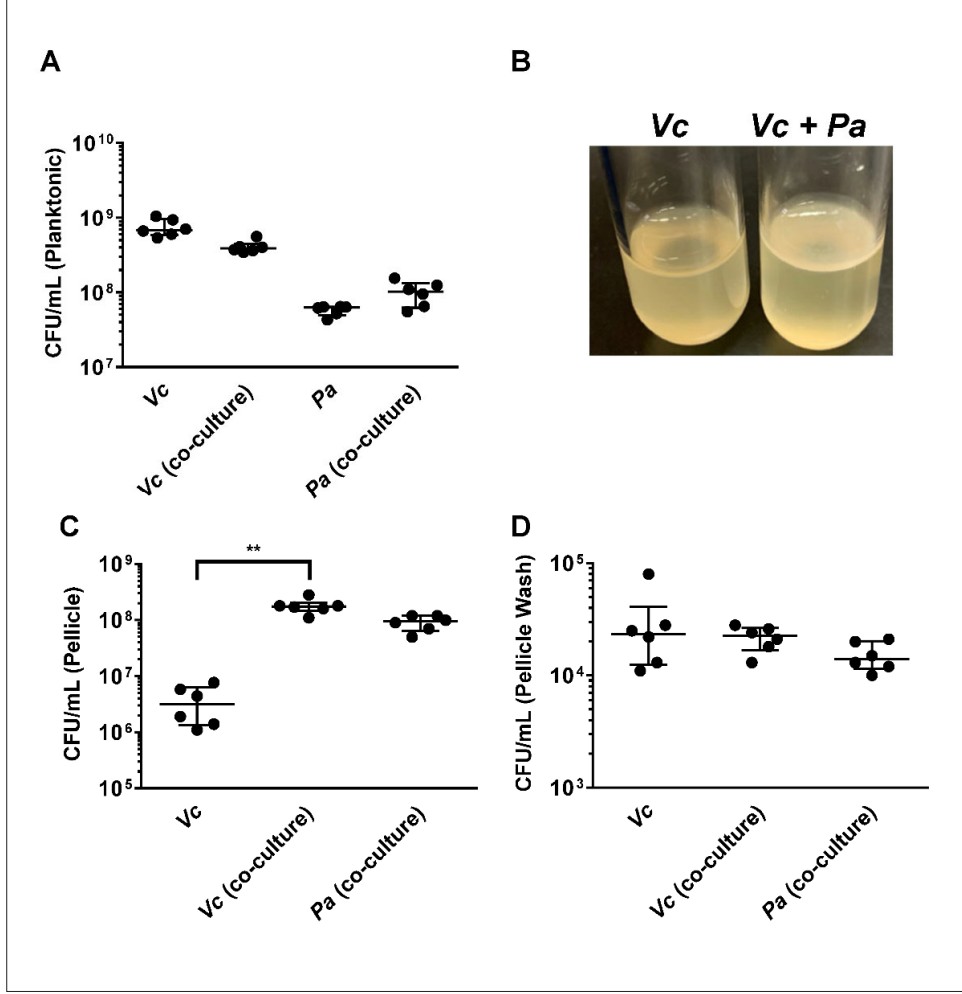

**Figure 3.** *Paracoccus aminovorans* (*Pa*) promotes biofilm formation of *Vibrio cholerae* (*Vc*). (**A**) Planktonic cell counts from cultures used for pellicle analysis of *Vc* and *Pa* grown together or in monoculture. (**B**) Representative images of pellicles formed by *Vc* grown in monoculture and co-culture with *Pa*. Colony-forming unit (CFU) counts of each strain in (**C**) pellicle samples and (**D**) spent medium used to wash the pellicle. Bars on graphs depict median value with 95% confidence interval (CI) and individual data points plotted. Unpaired nonparametric *U*-test (Mann–Whitney): **$p \leq 0.01$.

The online version of this article includes the following source data and figure supplement(s) for figure 3:

**Source data 1.** Viable cell count in for Paracoccus aminovorans (Pa) and *Vibrio cholerae* (Vc) in the planktonic phase in mono- and co-cultures.

**Source data 2.** Pellicle formation in Vc mono- and Vc/Pa coculture.

**Source data 3.** Viable cell count in for Paracoccus aminovorans (Pa) and *Vibrio cholerae* (Vc) in the pellicle formed by mono- and co-cultures.

**Source data 4.** Viable cell count in for Paracoccus aminovorans (Pa) and *Vibrio cholerae* (Vc) released by washing of the pellicle formed by mono- and co-cultures.

**Figure supplement 1.** *Paracoccus aminovorans* (*Pa*) cultures increase biofilm production in *Vibrio cholerae* (*Vc*).

**Figure supplement 1—source data 1.** Crystal violet assays of biofilm formed by Vc mono- and Vc/Pa co-cultures.

**Figure supplement 1—source data 2.** Crystal violet assays of biofilm formed by Vc mono- and Vc/Pa co-cultures using vpsL mutants with and without complementation plasmids.

**Figure supplement 1—source data 3.** Viable cell count of of pellicles formed by Vc mono- and Vc/Pa co-cultures using vpsL mutants with and without complementation plasmids.

growth and pellicle formation (i.e., biofilm formation at the air–liquid interface) of both species were monitored. There was a small difference (less than twofold) in growth in the planktonic phase of either *Vc* or *Pa* in the co-cultures when compared to the cultures containing a single species (*Figure 3A*). However, more importantly, the *Vc/Pa* co-culture formed a pellicle that was visibly thicker and more robust than that formed by *Vc* monoculture (*Figure 3B*). The *Pa* monoculture did not form a visible pellicle. The co-culture pellicle samples were carefully lifted and removed from the culture medium, washed, and agitated to release single cells for enumeration of each species. Compared to *Vc* monoculture, the co-culture samples contained over 50-fold more *Vc* cells while the ratio of *Vc* to *Pa* approached approximately 1:1 (*Figure 3C*). Moreover, only a small fraction (0.01%) of *Vc* and *Pa*

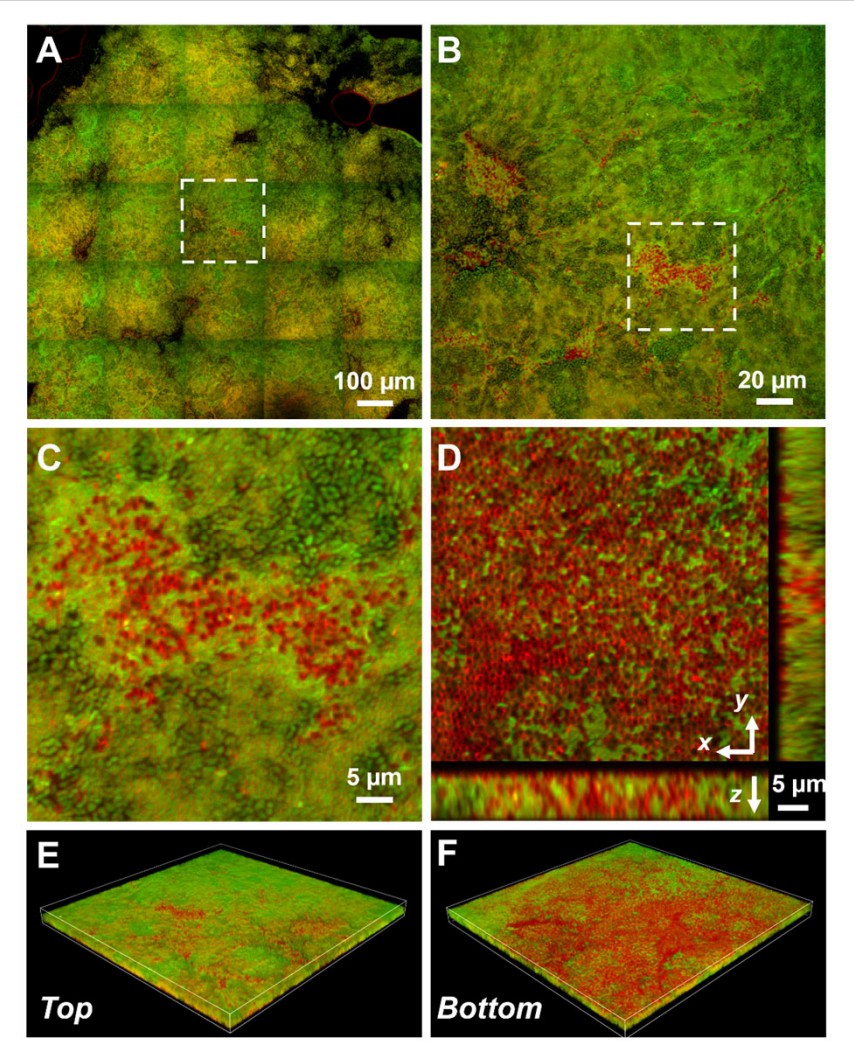

**Figure 4.** Representative microscopy images of *Vibrio cholerae* (*Vc*) and *Paracoccus aminovorans* (*Pa*) dual-species pellicles. (**A**) Large-scale cross-sectional image of the internal structure in a co-culture pellicle. All cells are stained with FM 4-64 and *Vc* cells constitutively express mNeonGreen. Therefore, the red signal in the overlay image corresponds primarily to *Pa* cells. The *Pa* cells can also be distinguished from *Vc* cells by their characteristic cocci shape. (**B**) Zoom-in view of the region highlighted in (**A**). (**C**) Zoom-in view of the region highlighted in (**B**). (**D**) Cross-sectional views of the region shown in (**C**), at the bottom of the pellicle. *Pa* cells exist mainly at the pellicle–liquid interface, with clusters of *Pa* cells penetrating into the interior of the pellicle. (**E, F**) Top (**E**) and bottom (**F**) views of the co-culture structure shown in (**B**), rendered in 3D.

The online version of this article includes the following figure supplement(s) for figure 4:

**Figure supplement 1.** Negative controls for wheat germ agglutinin (WGA) staining and the reporter strain.

could be washed off from the isolated pellicles (*Figure 3D*), suggesting that these species are tightly integrated into the pellicle.

Based on the above data, we hypothesize that *Vc* and *Pa* form dual-species biofilms at the air–liquid interface. This is unexpected because *Vc* is known to form a clonal community in both in vitro and in vivo biofilms and these are known to exclude other species, including even planktonic *Vc* cells (*Millet, 2014*; *Nadell et al., 2015*). To test this hypothesis, we transferred the co-culture pellicles onto coverslips for imaging with confocal microscopy (*Figure 4*). All cells in the pellicle were stained with FM 4-64 membrane dye, and *Vc* cells were differentiated from *Pa* using a constitutively produced mNeonGreen reporter (*Shaner et al., 2013*) expressed from a neutral *Vc* locus (*Dalia et al., 2014*). *Pa* cells have a cocci morphology that is distinct from the characteristic *Vc* curved-rod shape. In the *Vc/Pa* co-culture pellicles, we observed a continuous film structure spanning the entire pellicle (*Figure 4A*). Notably, cocci-shaped *Pa* cells were clearly visible in the co-culture pellicle (*Figure 4B and C*), consistent with the CFU quantification in *Figure 3A*. Interestingly, *Pa* cells were found throughout the pellicle, with a higher abundance in the bottom layer (*Figure 4D–F*) and in close association with *Vc* cells. In summary, we found that *Vc* and *Pa* coexist stably in the pellicle structure and this relationship may explain the mechanism by which *Pa* resists displacement in humans during active *Vc* infection.

Next, we used a standard crystal violet (CV) microtiter plate assay (*O'Toole, 2011*) to quantitatively evaluate how *Vc* and *Pa* interact under pellicle-forming conditions. *Vc* and *Pa* were simultaneously inoculated into the wells of microplates in two different *Vc:Pa* ratios (1:1 and 1:10). We also tested if the viability of *Pa* was crucial for this interaction by using heat-killed *Pa* as a control. Consistent with our pellicle compositional analysis, *Vc* formed a more robust biofilm than *Pa* under these conditions as demonstrated by increased CV staining in wells containing *Vc* only compared to wells containing *Pa* only (*Figure 5A*). Importantly, CV staining was increased in wells containing *Vc* and live *Pa* compared to wells with *Vc* only in a concentration-dependent manner (*Figure 5A*). In contrast, CV staining was not different in wells containing *Vc* and heat-killed *Pa* compared to wells with *Vc* only (*Figure 5A*).

To replicate our mouse experiments (*Figure 2*), we also tested if the order in which the two species encounter one another is critical for the *Vc* biofilm enhancement phenotype. *Pa* was grown in wells 24 hr before the addition of *Vc*. As in our previous results in the co-inoculation experiment, an increase in CV staining was observed in wells in which the two species were added sequentially, but not in the wells with *Vc* only (*Figure 3—figure supplement 1*). Moreover, wells pre-incubated with heat-killed *Pa* and subsequently inoculated with *Vc* had no increase in CV staining compared to wells inoculated with *Vc* alone (*Figure 3—figure supplement 1*). Together, our biofilm quantification data suggest that the presence of *Pa*, regardless of the order of encounter, results in an enhanced biofilm formation of *Vc*.

## *Vibrio* exopolysaccharide is essential for a stable biofilm structure formed by *Vc* and *Pa*

To understand what biofilm component is required for the enhancement of biofilm production in *Vc/Pa* co-culture, we repeated the above experiments with a Δ*vpsL* *Vc* mutant that cannot produce the *Vibrio* exopolysaccharide (VPS) necessary for mature biofilm formation (*Yildiz and Schoolnik, 1999*). In contrast to our observations in culture of a *vpsL*$^+$ strain, there was no significant increase in CV staining in wells with Δ*vpsL* mutant and *Pa* co-culture when compared to wells with the Δ*vpsL* mutants only (*Figure 5B*). The biofilm formation defects of the *Vc* Δ*vpsL* mutants and the increased biofilm formation response in the presence of *Pa* could be restored by the introduction of a plasmid constitutively expressing *vpsL* (*Figure 3—figure supplement 1*).

We then tested if the presence of *Pa* changes *vps* gene expression in *Vc* as one of the mechanisms responsible for enhanced biofilm formation. *Vc* monoculture and *Vc/Pa* co-culture were grown statically to induce pellicle formation, and cells near the air–liquid interface where pellicle was formed were collected. Using qRT-PCR, we determined that the relative transcript levels of *vpsL* (the first and a representative gene in the *vpsII* operon in *Vc*) were significantly higher in the *Vc/Pa* co-culture than that in the *Vc* monoculture after 24–48 hr of growth (*Figure 5C*). There was no significant difference in *vpsL* transcript levels after 72 hr of co-culture. Our results suggest that *Pa* induces *Vc* biofilm gene expression when the two species are cultured together, especially at the early stage of interaction, creating the conditions for increased pellicle formation in co-cultures.

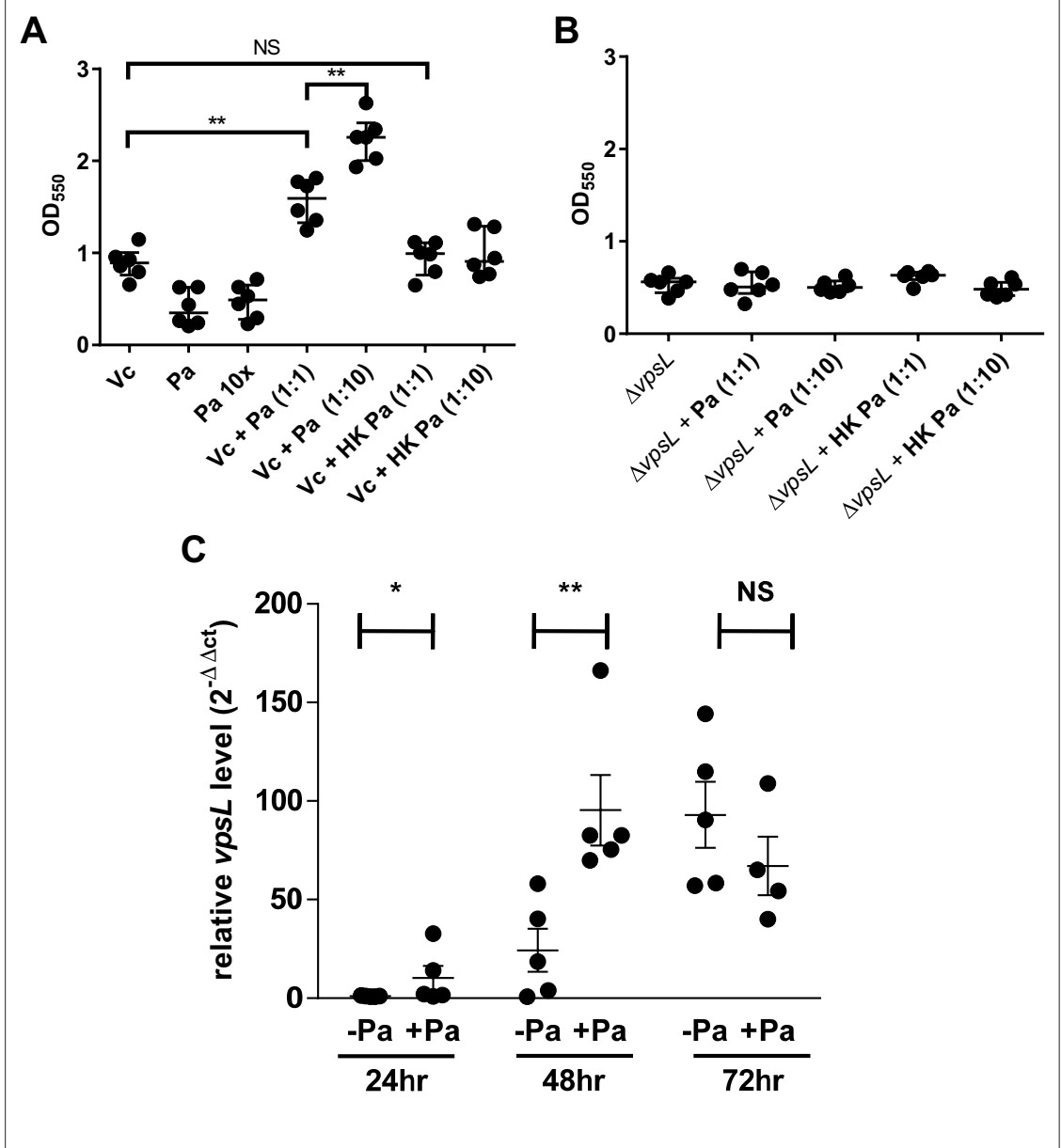

**Figure 5.** *Paracoccus aminovorans* (*Pa*) increases biofilm production in *Vibrio cholerae* (*Vc*). Crystal violet assays were performed in 96-well microtiter plates to quantify biofilm formation. Overnight-grown (**A**) wild-type *Vc* or (**B**) *ΔvpsL* mutant and *Pa* cultures were diluted to a final concentration of $10^6$ colony-forming unit (CFU) in a total volume of 200 µL/well. In samples containing a 1:10 ratio of *Vc/Pa*, *Pa* was diluted to a final concentration of $10^7$ CFU. Samples with heat-killed (HK) *Pa* are specified on the x-axis. Microtiter plates were incubated at 37°C for 24 hr. Crystal violet staining and ethanol solubilization were performed as previously described (*O'Toole, 2011*). Absorbance of the crystal violet stain was measured at 550 nm using a BioTek Synergy HTX plate reader. Data are represented with horizontal lines indicating the mean with standard deviation. Unpaired *U*-test (Mann–Whitney); **p≤0.01. (**C**) *Vc* and *Pa* were co-cultured at a 1:10 ratio statically for 72 hr at 30°C. At the specified time points, culture from the air–liquid interface was sampled and the RNA extracted. Relative *vpsL* transcript levels were determined by qRT-PCR using housekeeping gene *groEL* and the ΔΔCT analytic method. Bars on graphs depict mean with standard error of mean (SEM). Mann–Whitney *U*-test was performed, **p<0.01, *p<0.05, NS p>0.05.

The online version of this article includes the following source data for figure 5:

**Source data 1.** Crystal violet assays of biofilm formed by Vc mono- and Vc/Pa co-cultures using WT with live or heat-killed Pa.

**Source data 2.** Crystal violet staining of biofilm formed by Vc mono- and Vc/Pa co-cultures using vpsL mutants with live and heat-killed Pa.

**Source data 3.** qRT-PCR analysis of vpsL gene in cells in biofilm formed by Vc mono- and Vc/Pa co-cultures.

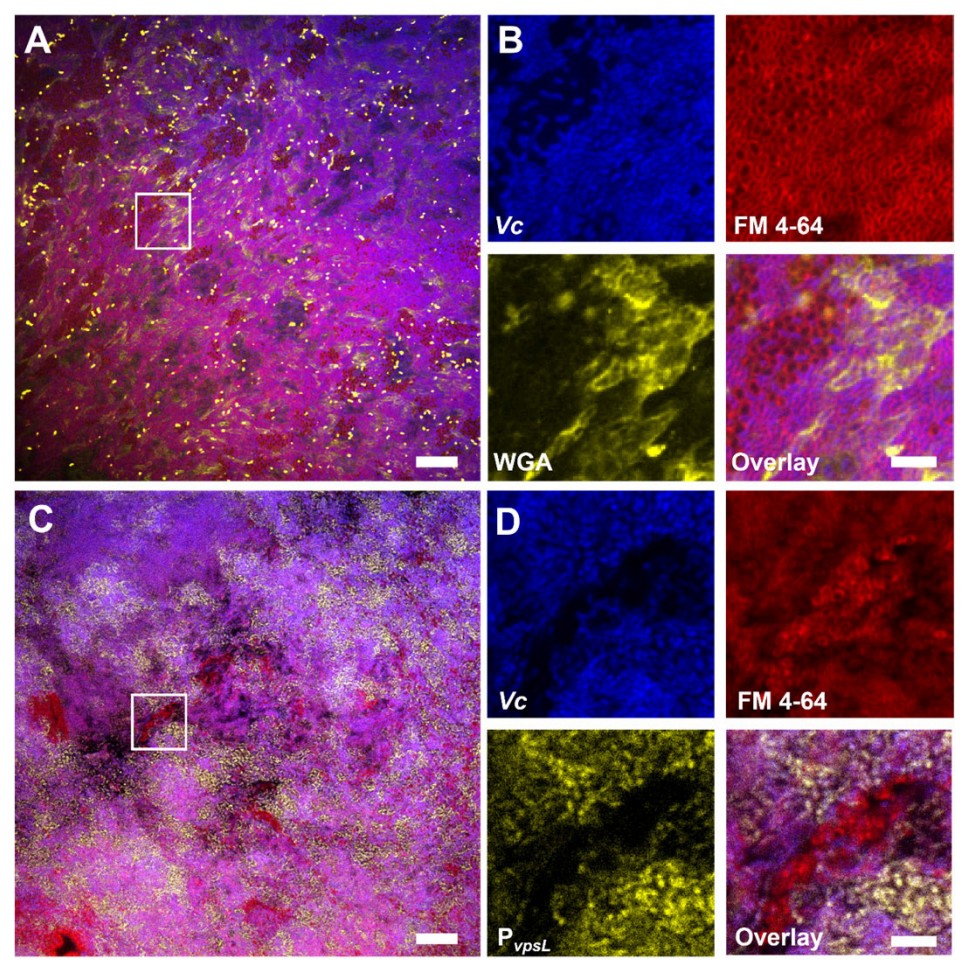

**Figure 6.** *Vibrio cholerae/Paracoccus aminovorans* (*Vc/Pa*) co-culture biofilms depend on *Vibrio* exopolysaccharide (VPS). (**A**) Representative cross-sectional view of the bottom layer of a co-culture pellicle with wheat germ agglutinin (WGA) staining. *Vc* cells constitutively express SCFP3A cytosolically; all cells were stained with FM 4-64 membrane stain; WGA is conjugated to Oregon Green and shown in yellow. Note that WGA also stained dead cells with an exposed peptidoglycan layer, corresponding to the bright spots in the image. Scale bar: 20 μm. (**B**) Zoom-in view of the region highlighted by the white box in (**A**). Shown are separate channels from *Vc* cell fluorescence (SCFP3A, blue), membrane staining (FM 4-64, red), WGA staining (Oregon Green, yellow), and the overlay of the three channels. *Pa* cells can be distinguished from *Vc* cells by both the absence of SCFP3A fluorescence and the distinct cell shape. Scale bar: 5 μm. (**C**) Representative cross-sectional view of a co-culture pellicle in which the *Vc* cells harbor a $P_{vpsL}$-*mNeonGreen* reporter. Scale bar: 20 μm. (**D**) Zoom-in view of the region highlighted by the white box in (**C**). Shown are separate channels from *Vc* cell fluorescence (SCFP3A, blue), membrane staining (FM 4-64, red), *vpsL* reporter (mNeonGreen, yellow), and the overlay of the three channels. Scale bar: 5 μm.

To further investigate the role of VPS in promoting co-culture biofilms, we stained the co-culture pellicle in situ with wheat germ agglutinin (WGA), a common stain for the N-acetylglucosamine (GlcNAc) moieties that is a component of VPS (***Yildiz et al., 2014***). Note that for Gram-negative bacteria, including *Vc* and *Pa*, the WGA lectin molecules do not pass through the outer membrane of healthy cells. Indeed, control experiments show that healthy *Pa* cells and *Vc* Δ*vpsL* mutant cells do not show positive VPS staining (***Figure 4—figure supplement 1***). To avoid spatial overlap with the membrane stain (excited at 561 nm), the *Vc* cells used in this experiment express a cyan-fluorescent protein SCFP3A (excited at 445 nm), and the WGA is conjugated to Oregon Green (excited at 488 nm). ***Figure 6A*** shows a *Vc/Pa* co-culture pellicle, in which extensive WGA signal is observed. Zoom-in view of the pellicle shows the characteristic envelope structures around clusters of *Vc* cells (***Figure 6B***), consistent with the known VPS staining pattern in submerged *Vc* biofilms

(*Berk et al., 2012*). We also noticed that regions with strong VPS signal are not necessarily spatially correlated with clusters of *Pa* cells.

Next, to demonstrate that co-culturing with *Pa* increases *vps* gene expression, we used a *Vc* strain harboring a P*vpsL*-*mNeonGreen* reporter and repeated the imaging of the co-cultured pellicle. *Figure 6C* shows that in the co-cultured pellicle *vspL* expression is elevated in subpopulations of *Vc* cells in the co-culture. Zoom-in views show that even adjacent to *Pa* cell clusters some *Vc* cells have high *vspL* expression and others have basal-level expression (*Figure 6D*, *Figure 4—figure supplement 1*). These results are consistent with the qRT-PCR results in *Figure 5C*, suggesting that *Pa* induces *Vc* biofilm gene expression in co-culture. Together, these results suggest that the physical presence of *Pa* in the co-culture pellicle augments the production of VPS in *Vc* cells, leading to increased *Vc* biofilm formation; the *Pa* cells, in turn, require VPS to be integrated into the 3D structure of the pellicle.

## Enhancement of *Vc* host colonization by *Pa* depends on biofilm exopolysaccharide

Biofilm-grown *Vc* cells are known to be more infectious in humans due to increased resistance to gastric pH and higher expression of virulence factors (e.g., such as the toxin co-regulated pilus, which mediates host colonization) compared to planktonically grown cells (*Tamayo et al., 2010*; *Gallego-Hernandez et al., 2020*; *Zhu and Mekalanos, 2003*). We hypothesize that because *Vc* biofilm formation is enhanced in the presence of *Pa* this results in increased virulence inside the host in a VPS-dependent manner. Previous studies report that *Vc* El Tor biotype strains require certain growth conditions (e.g., AKI) to induce virulence gene expression in vitro (*DiRita et al., 1996*; *Iwanaga et al., 1986*). However, these specific growth conditions include culture agitation, which would prevent proper interaction of *Vc* and *Pa*. Therefore, using qRT-PCR, we measured virulence gene expression in *Vc/Pa* co-culture and compared this to *Vc* monoculture grown statically in identical conditions to those that favor biofilm formation. As expected, *ctxA* and *tcpA* relative transcript levels were low in all samples; however, there was a modest increase in the expression of these two virulence genes in the *Vc/Pa* co-culture compared to the *Vc* monoculture (*Figure 7—figure supplement 1*).

To further test our hypothesis and measure if the effect of the *Vc/Pa* biofilm interaction impacts host colonization, we compared the colonization efficiency between wild-type (WT) or the Δ*vpsL* mutants in infant mice with and without *Pa* pre-colonization. As shown previously (*Fong et al., 2010*), the Δ*vpsL* mutant was able to colonize the mouse small intestine as well as the WT *vpsL*⁺ strain, confirming that VPS is not absolutely required for host colonization when *Vc* was administered to the animals alone. In contrast, while *Pa* increased WT *vpsL*⁺ *Vc* colonization, the Δ*vpsL* mutant did not exhibit the enhanced colonization phenotype in the *Pa* pre-colonized mice (*Figure 7A*). Host colonization enhancement caused by *Pa* pre-colonization could be restored in the Δ*vpsL* mutants with a plasmid constitutively expressing *vpsL,* but not with the empty vector (*Figure 7—figure supplement 2*). Similar results were observed using the co-infection model; when *Vc* Δ*vpsL* mutants were coinfected with *Pa*, there was no increase in host colonization (*Figure 7B*). Together, we concluded that the enhancement of *Vc* intestinal colonization in the presence *Pa* is dependent on the VPS, consistent with our in vitro data.

## Accessory biofilm matrix proteins are involved in *Pa* and *Vc* interaction

Mature *Vc* biofilm is stabilized with a variety of accessory matrix proteins in addition to the VPS (*Berk et al., 2012*; *Fong et al., 2006*; *Fong and Yildiz, 2007*). To interrogate the roles of these components in the interactions between *Vc* and *Pa*, we tested mutants lacking the cell-cell adhesion proteins RbmA (*Berk et al., 2012*; *Fong et al., 2006*; *Absalon et al., 2011*) and mutants lacking surface adhesion redundantly conferred by RbmC and Bap1 (*Berk et al., 2012*; *Fong and Yildiz, 2007*; *Absalon et al., 2011*) for their ability to increase biofilm formation in the presence of *Pa* using CV assays (*Figure 7C*). When compared to the wells containing the Δ*rbmA* mutant alone, the CV staining was higher in the wells with the Δ*rbmA* mutant co-cultured with *Pa*. However, the increase was not as high in the Δ*rbmA* mutant/*Pa* co-culture compared to that in WT *Vc/Pa* co-culture (*Figure 7C*). Furthermore, the presence of *Pa* did not increase CV staining in the wells containing the Δ*rbmC* Δ*bap1* mutants (*Figure 7C*) because the Δ*rbmC* Δ*bap1* mutant is not able to adhere to the interface.

We then performed the infant mouse colonization experiments with *Vc* biofilm matrix protein mutants using the *Pa* co-infection model to test the roles of these proteins in vivo. For this series of experiments, each *Vc* biofilm mutant was coinfected into the animals with *Pa* in 1:1 ratio. In agreement

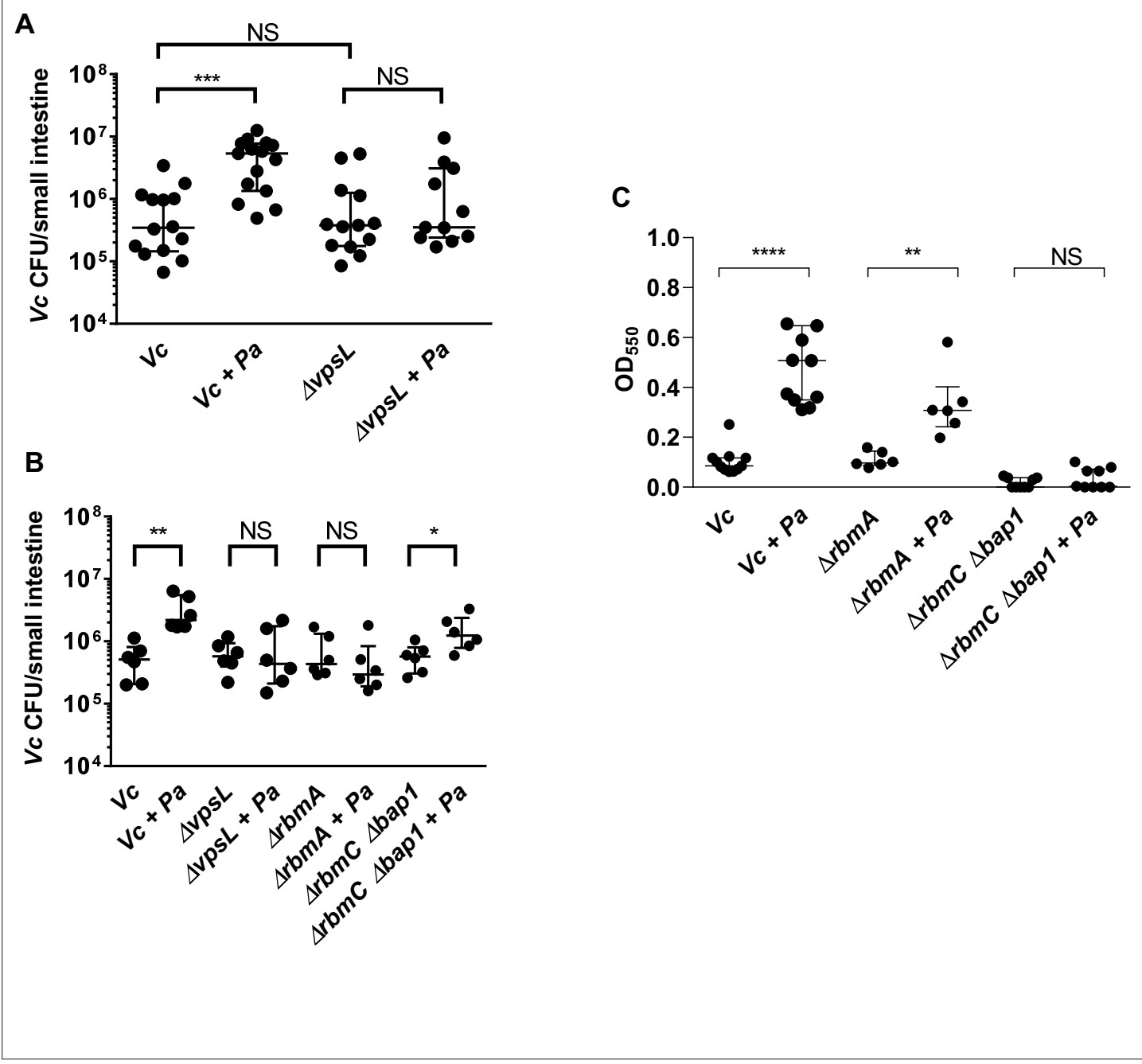

**Figure 7.** Enhanced *Vibrio cholerae* (*Vc*) intestinal colonization in the presence of *Paracoccus aminovorans* (*Pa*) is dependent on *Vibrio* exopolysaccharide (VPS) and accessory matrix proteins. (**A**) 3-day-old infant mice were intragastrically inoculated with LB or $10^7$ colony-forming unit (CFU) of *Pa* every 12 hr for a period of 48 hr, and subsequently infected with $10^6$ CFU of a *Vc* strain defective for extracellular matrix production (*ΔvpsL*). Mice were sacrificed 20–24 hr post infection, and small intestine samples were taken to enumerate *Vc* CFU. Data from infection with the wild-type *Vc* strain (*Figure 2B*) are shown again here for comparison purposes. (**B**) *Vc* WT or different biofilm mutants were mixed with *Pa* in 1:1 ratio, and the mixture was used immediately for animal infection. Mice were sacrificed 20–24 hr post infection, and small intestine samples were taken to enumerate *Vc* CFU. Each symbol represents an individual mouse. Data from infection with the WT *Vc* strain (*Figure 2C*) are shown again here for comparison purposes. (**C**) Crystal violet assays performed in 96-well plates to quantify pellicle formation. Overnight cultures of *Vc ΔrbmA, ΔrbmC Δbap1,* and *Pa* were diluted in fresh LB and plated as 200 µL/well. Samples were co-cultured in 1:10 ratios of Vc/Pa and incubated at 37°C for 24 hr. Crystal violet staining was then performed as previously described. For all panels, horizontal lines indicating median with standard deviation are shown. Unpaired Mann–Whitney *U*-test; ****p≤0.0001, ***p≤0.001, **p≤0.01, *p≤0.05, NS p>0.05.

The online version of this article includes the following source data and figure supplement(s) for figure 7:

**Source data 1.** Viable cell count of Vc in the mouse intestine with and without Pa pre-colonization using Vc WT and vpsL mutants.

Figure 7 continued

**Source data 2.** Viable cell count of Vc in the mouse intestine with and without Pa pre-colonization using Vc WT and different biofilm matrix protein mutants.

**Source data 3.** Crystal violet staining of biofilm formed by different biofilm matrix protein mutants with and without Pa.

**Figure supplement 1.** *ctxA* and *tcpA* gene expression in *Vibrio cholerae* (*Vc*) monoculture and *Vibrio cholerae-Paracoccus aminovorans* (*Vc-Pa*) co-culture.

**Figure supplement 1—source data 1.** qRT-PCR analysis of virulence genes in Vc with and without Pa.

**Figure supplement 2.** Complementation of *vpsL* mutant in vivo.

**Figure supplement 2—source data 1.** Viable cell count of Vc in mouse intestine with and without Pa pre-colonization using vpsL mutant with and without vpsL complementation plasmid.

with our in vitro results, while host colonization was significantly higher for WT *Vc* coinfected with *Pa* than without *Pa* (**Figure 2C**), ΔrbmA mutants did not show any increase in host colonization when coinfected with *Pa*, and ΔrbmC Δbap1 mutants demonstrated increased colonization with *Pa* but less colonization compared with WT *Vc* co-infection with *Pa* (**Figure 7B**). These results indicate that the ability of *Vc* to form a structurally intact biofilm is important for the enhancement of colonization facilitated by the presence of *Pa*.

## Discussion

Evidence that the composition of the gut microbiota influences the clinical outcomes of enteric infections in humans is accumulating (**Ubeda et al., 2017**; **Weil et al., 2019**). Several studies have identified commensal species and underlying colonization resistance mechanisms that could be protective against *Vc* infection. While these studies suggest that microbiota species reduce *Vc* virulence through various mechanisms during the early stages of infection (**Alavi et al., 2020**; **Hsiao et al., 2014**; **You et al., 2019**), the precise role of these colonization resistance mechanisms in impacting susceptibility to cholera in humans has only begun to be appreciated. For instance, a gut bacterium in the genus *Blautia* was recently found to encode functions that confer colonization resistance (e.g., bile salt hydrolase) to *Vc* infection (**Alavi et al., 2020**). Consistent with this finding, our previous stool microbiome study also independently identified that a species in the genus *Blautia* is correlated with decreased susceptibility to *Vc* infection (**Levade et al., 2020**; **Midani et al., 2018**).

While previous studies have identified microbiota-associated mechanisms that are protective against *Vc* infection, examples of interactions between *Vc* and a human-associated microbiota species that increases *Vc* pathogenicity are scarce. Although *Escherichia coli* and *Vc* are believed to reside in different intestinal niches, one previous study showed that an atypical *E. coli* isolated from a mouse that does not ferment lactose can increase the virulence of a quorum-sensing (QS)-defective *Vc* strain N16961 (**Yoon et al., 2016**). How QS-proficient *Vc* strains, which are prevalent among toxigenic clinical isolates (**Wang et al., 2011**), respond to *E. coli* in the human gut remains to be studied. In contrast, a recent study showed that *E. coli* motility facilitates aggregation of these two organisms in a dual-species biofilm, but there was no impact of such aggregation on *Vc* intestinal colonization (**Wang et al., 2021**). Indeed, coaggregation between *Vc* and other microbiota species has been observed (**Toh et al., 2019**), but these associations are not known to have a direct influence on *Vc* pathogenicity. This is consistent with our prior human studies in which *E. coli* species were present in the gut microbiota of persons during active *Vc* infection, but these were not correlated with active *Vc* infection (**Midani et al., 2018**). Our findings highlight the importance of coupling mechanistic studies (in vitro and animal models) with human microbiome data analysis to pinpoint the relevant species and interactions involved in enteric infections.

Here, we show that the presence of a human gut microbe *Pa* promotes *Vc* host colonization, which is consistent with our prior human study in which *Pa* was more likely to be present in persons infected with *Vc*. This raises the possibility that uncharacterized interactions between *Vc* and members of the gut microbiota may exacerbate *Vc* virulence and contribute to increased morbidity. While our previous study showed that *Vc* has a better growth yield in the conditioned medium (CM) harvested from stationary culture of *Pa* than in the CM harvested from stationary culture *Vc*, the growth yield of *Vc* in the CM harvested from stationary culture of *Pa* does not reach as high as *Vc* in fresh medium (**Midani**

*et al., 2018*). These results suggest that *Pa* does not actively secrete any molecules that enhance *Vc* growth. Most likely, the limited growth yield increase of *Vc* in the CM harvested from *Pa* is due to the presence of a trace amount of nutrients that have been exhausted in the CM harvested from *Vc*. In contrast, this study establishes a plausible mechanism used by *Pa*, and perhaps other gut microbes, to increase the virulence of *Vc* through induction of biofilm formation, a physiological state in which *Vc* is known to increase expression of other virulence factors critical for human infection and disease (*Tamayo et al., 2010*; *Gallego-Hernandez et al., 2020*). *Vc* biofilms have also been demonstrated to deform and even damage tissue-engineered soft epithelia mimicking the host tissue (*Cont et al., 2020*), suggesting that in vivo-formed biofilm structures could negatively impact host gut physiology.

While VPS and other biofilm components are not usually considered critical host colonization factors, we found that these macromolecular structures were essential for the enhancement of *Vc* host colonization induced by *Pa*. Whether these components mediate other *Vc*-gut microbe interaction has not been studied. Interestingly, many gut microbes appear to exist in the form of mixed-species biofilms on mucosal surfaces (*Sadiq et al., 2021*), suggesting that microbiota-induced biofilm enhancement could play a major role in modulating virulence of other pathogens. Many structural components, regulatory factors, and signaling transduction pathways that control biofilm formation in *Vc* have been well characterized (*Teschler et al., 2015*), and these factors could be targeted for manipulation by other gut microbes that modulate *Vc* virulence. For example, 3,5-dimethylpyrazin-2-ol (DPO) was recently discovered as a new class of *Vc* QS autoinducer that binds to the transcription factor VqmA to activate expression of *vqmR*, which encodes a small regulatory RNA that downregulates *Vc* biofilm formation. The VqmA-VqmR system can be activated both in vitro by *E. coli* and in vivo by *B. obeum* (*Hsiao et al., 2014*; *Papenfort et al., 2017*), and results in suppression of biofilm formation. Interestingly, *Pa* demonstrates the opposite tendency by promoting *Vc* biofilm formation, with implications for the enhancement of *Vc* colonization, in contrast to other commensal bacteria.

Many aspects of the *Vc/Pa* interaction are still unclear. What is the selective advantage that fosters the formation of dual-species biofilm? Investigation of the structure–function relationship in other multispecies biofilms, such as dental biofilms, demonstrates a coordinated organization of each species that allows for optimal nutrient and oxygen usage, as well as mechanical stability (*Christensen et al., 2002*; *Mark Welch et al., 2016*). While we did not observe any growth yield enhancement in the planktonic phase of the co-culture, there was a significant increase of *Vc* and *Pa* abundance in the co-culture pellicle at the air–liquid interface. Thus, a possible driving force of this interaction could be the optimization of nutrient sharing and distribution, or removal of toxic metabolites accumulated during growth. Biofilm formation also changes the biophysical properties of the microbial community that could facilitate host adhesion (*Jiang et al., 2021*). The exact mechanism used by these two species to detect and coordinate with each other remains unclear. Secreted small molecules produced by *Pa* do not appear to impact *Vc* as evidenced by our prior studies in which *Vc* cultured in *Pa* spent-cell supernatant did not result in increased biofilm formation (*Midani et al., 2018*). Therefore, we surmised that the close physical association between *Vc* and *Pa* cells in space in the co-culture pellicles is required for the enhanced biofilm formation. This hypothesis is supported by our microscopy analysis. The *Vc–Pa* interaction has two reciprocal aspects: first, *Pa* activates the production of VPS in *Vc* cells, leading to enhanced pellicle formation. Future characterizations of *Pa* could potentially elucidate the underlying molecular mechanism of this effect *Pa* has on *Vc*. Second, in order to be integrated into the pellicle structure, *Pa* cells seem to physically adhere to *Vc* cells or alternatively to the extracellular matrix that *Vc* cells secrete. Future biochemical and biophysical studies to investigate this relationship may provide new insights about the interactions between *Pa* and *Vc* biofilm, and about pathogen–gut microbe interactions in general. Other members of the *Paracoccus* genus are known to form biofilms and encode adhesins to facilitate surface attachment (*Yoshida et al., 2017*; *Srinandan et al., 2010*), and the potential role of these adhesins in facilitating interaction with *Vc* remains to be studied.

In addition to VPS, the main structural component of *Vc* biofilms, we have shown that matrix proteins, including RbmA, Bap1, and RbmC, are also critical for the enhancement of *Vc* biofilm by *Pa* in the neonatal mouse colonization model. Specifically, we showed that cell–cell adhesion, conferred by RbmA, is more important than cell–surface adhesion conferred by RbmC/Bap1. Previous work established that RbmA is responsible for maintaining dense, structurally robust biofilms (*Berk et al., 2012*; *Absalon et al., 2011*; *Fong et al., 2017*; *Yan et al., 2016*). Therefore, we contemplate that the Δ*rbmA* mutant biofilm may be mechanically impaired and become destroyed in the gut environment

by physical forces such as peristaltic flow, food particle collision, and others. Likewise, such physical forces could impede biofilm colonization when biofilm adhesins are absent, which explains the slight defect in colonization of the ΔrbmCΔbap1 mutant. Indeed, recently *Vc* cells have been visualized in the mouse gut and the biofilm population has been shown to be located primarily at the tip of the villi, where fluid shear is strongest (*Gallego-Hernandez et al., 2020*). These relationships may be further revealed by additional imaging of the distribution of the *Vc* and *Pa* cells and potentially other commensal bacteria in the new mouse model presented here.

In conclusion, we describe a novel interaction between *Vc* and a gut microbe found in high abundance in *Vc*-infected persons that leads to a significant change in *Vc* biofilm behaviors, as well as an increase in the virulence of the pathogen. Our findings are also consistent with other observations that rare gut microbial species can have significant impacts on microbial ecosystems (*Jousset et al., 2017*). This study adds to the growing number of pathogen–gut microbial species interactions that may impact outcomes in human diseases.

## Materials and methods
### Prior published study sample collection and analysis
In a prior study, we enrolled household contacts of persons hospitalized with cholera at the International Centre for Diarrheal Disease Research, Bangladesh (icddr,b; *Midani et al., 2018*). Briefly, in this previously published study, household contacts were followed prospectively with rectal swab sampling, 30 days of clinical symptom report, and vibriocidal titer measurements, and 16S rRNA sequencing was performed on rectal swab sampling from the day of enrollment in the study (*Midani et al., 2018*). Persons with evidence of *Vc* infection at the time of enrollment in the study were compared to those who did not have evidence of infection in a model to detect gut microbes that were differentially abundant during *Vc* infection (*Midani et al., 2018*). *Vc* infection was defined as *Vc* DNA identified on 16S rRNA sequencing or a positive *Vc* stool culture. In this previously published study, we used a machine learning method called a support vector machine (SVM), which utilizes patterns of OTU relative abundance to detect OTUs associated with infected compared to uninfected persons. This SVM was used with a recursive feature elimination algorithm that simplifies models and increases accuracy of the identification of differentially associated OTUs by removing uninformative bacterial taxa (*Midani et al., 2018*). For this study, we reexamined the microbiome data from household contacts at the time of enrollment to quantify the abundance of 16S rRNA reads that mapped to *Pa* OTUs between uninfected study participants and infected participants.

### Strains and culture conditions
All *Vc* strains used in this study are streptomycin-resistant derivatives of C6706, a 1991 El Tor O1 clinical isolate from Peru (*Thelin and Taylor, 1996*). As *luxO* mutations are readily isolated in this strain (*Jung et al., 2015*; *Stutzmann and Blokesch, 2016*), DNA sequencing or cell-density-dependent $P_{qrr4}$-*lux* reporter assay (*Jung et al., 2015*) was performed to confirm our strains carry a functional WT allele of *luxO*. The in-frame Δ*vpsL* deletion mutants used in various assays were previously described (*Waters et al., 2008*). The Δ*rbmA* and Δ*rbmC*Δ*bap1* mutants were constructed by allelic exchange (*Skorupski and Taylor, 1996*) using specific suicide vectors described before (*Nadell et al., 2015*; *Yan et al., 2016*). *Vc* strains used for microscopy experiments, Δ*vc1807::P_{tac}–mNeonGreen*, Δ*vc1807::P_{tac}–SCFP3A-spec^R*, and Δ*vc1807::P_{tac}–SCFP3A-spec^R* Δ*LacIZ::P_{vpsL}-mNeonGreen*, were constructed using natural transformation as previous described (*Dalia et al., 2014*). For *vpsL* complementation, the reading frame of *vpsL* was first amplified by PCR using Phusion DNA polymerase, C6706 genomic DNA, and primers gataacaatttcacaatgaaggaaaaaagcagaatacgcattac and gaattctgtttcctgttaatacgcgttt tttccaacaaatcctttg. Vector pMMB67eh was linearized by PCR using primers tgtgaaattgttatccgctc and caggaaacagaattcgag. Gibson assembly was used to fuse the two fragments together. The resulting plasmid was transformed into DH5α and confirmed by sequencing. This plasmid was introduced into *Vc* by triparental mating with a helper plasmid pRK2013. The *Pa* used in our experiments is a Strep^R isolate derived from the ATCC-type strain (ATCC #49632). *Vc* and *Pa* overnight cultures were grown with aeration in LB at 30°C. Heat-killed strains were incubated at 60°C for 2 hr prior to experimentation. Unless specified, media were supplemented with streptomycin (Sm, 100 μg/mL) and chloramphenicol (Cm, 10 μg/mL) when appropriate.

## Animal studies

For establishing colonization of the microbiota species, 3-day-old suckling CD-1 mice (Charles River Laboratories) were fasted for 1 hr, then orally dosed with *Pa* at a concentration of $10^7$ CFU using 30-gauge plastic tubing, after which the animals were placed with a lactating dam for 10–12 hr and monitored in accordance with the regulations of Tufts Comparative Medicine Services. This inoculation scheme was followed an additional three times at 12, 24, and 36 hr. 12 hr after the last inoculation (i.e., 48 hr after the first inoculation), mice were infected with $10^6$ CFU of *Vc*, WT C6706 or mutant strain, or LB as a vehicle control in a gavage volume of 50 µL to evaluate the effect of *Pa* pre-colonization on *Vc* host colonization. At 18–24 hr post infection, animals were sacrificed, and small intestine tissue samples were collected and homogenized for CFU enumeration. WT *Vc* is *lac*+ and appears blue on medium containing X-gal while *Pa* appears white on the same medium. For co-infection experiments, cultures of *Vc* and *Pa* strains were mixed in a 1:1 ratio and mice were orally dosed with a final bacterial count of $10^6$ CFU. Mice were sacrificed 20–24 hr post infection, and small intestine samples were processed as outlined above to evaluate the colonization efficiency of both species.

## Ethics statement

All animal experiments were performed at and in accordance with the rules of the Tufts Comparative Medicine Services (CMS), following the guidelines of the American Veterinary Medical Association (AVMA), as well as the Guide for the Care and Use of Laboratory Animals of the National Institutes of Health. All procedures were performed with approval of the Tufts University CMS (protocol# B 2018-99 and #B2021-81). Euthanasia was performed in accordance with the guidelines provided by the AVMA and was approved by the Tufts CMS. The previously published study from which *Figure 1* is derived (*Midani et al., 2018*) received approval from the Ethical Review Committee at the icddr,b and the institutional review boards of Massachusetts General Hospital and the University of Washington, and in that study participants or their guardians provided written informed consent.

## Pellicle composition analysis

To assess pellicle composition, overnight cultures of *Vc* and *Pa* were inoculated into glass culture tubes (18 × 150 mm) containing 2 mL LB media in a ratio of 1:10 *Vc* ($10^6$) to *Pa* ($10^7$) CFU, and co-cultures were allowed to grow statically at room temperature for 3 days. Following static growth, floating pellicles were carefully transferred into sterile 1.5 mL Eppendorf tubes containing 1 mL LB, and samples were gently spun down to wash away any planktonic bacteria. Planktonic cells were removed, and cell pellets of pellicle samples were resuspended in 1 mL of fresh LB media. All samples, including supernatant from the pellicle wash step, were serially diluted and plated on Sm/X-Gal media to differentiate *Vc* (blue) and *Pa* (white) colonies.

## Crystal violet biomass assays

CV biofilm assays were performed as described previously in 96-well flat-bottom clear, tissue-culture-treated polystyrene microplates (Thermo Fisher; *O'Toole, 2011*). In each well, *Vc* ($10^6$ CFUs) and/or *Pa* ($10^6$ or $10^7$ CFUs) were inoculated into 200 µL of medium. Plates were then sealed using a gas-permeable sealing film (BrandTech) and incubated at 37°C. Planktonic culture was removed after 24 hr of incubation, and plates were washed with distilled water once. Attached biofilms were stained with 0.1% CV at room temperature for 15–20 min. The amount of biomass adhered to the sides of each well was quantified by dissolving the CV in 95% ethanol, and the absorbance of the resulting solution was measured at 550 nm using a plate reader.

## Microscopy

Liquid LB culture of *Vc*, *Pa*, and co-cultures (*Vc:Pa* = 1:10) were prepared according to the procedures described above. To image pellicles, we used a modified literature procedure (*Fiebig, 2019*). Monocultures and co-culture pellicles were first prepared following the procedure described above, except that 3 mL of the culture was incubated in a 5 mL culture tube. After 3 days of incubation at room temperature, the pellicles were carefully picked up by the large end of a 200 µL pipette tip, transferred to a coverslip (22 × 60 mm, no. 1.5), and immediately covered with another square coverslip to prevent drying. The LB medium contained 4 µg/mL FM 4-64 stain (Thermo Fisher) to stain all cells. To stain VPS, the LB medium additionally contained 4 µg/mL of WGA conjugated to Oregon

Green (Thermo Fisher). The stained biofilms were imaged with a Nikon-W1 confocal microscope using ×60 water objective (numerical aperture = 1.20). The imaging window was 222 × 222 μm². For large-scale view, a 5 × 5 tiling was performed. For zoom-in view, the z-step size was 0.5 μm and the pixel size was 108 nm. For large-scale view, the z-step size was 1 μm and the pixel size was 216 nm. The mNeonGreen (or SCFP3A) expressed by Vc was imaged at 488 nm (or 445 nm) excitation, FM 4-64 at 561 nm, and WGA-Oregon Green at 488 nm with the corresponding filters. All presented images are raw images processed from Nikon Element software.

## Quantitative-real-time PCR (qRT-PCR) for *Vc* virulence factors

Liquid LB culture of *Vc*, *Pa*, and co-cultures (*Vc:Pa* = 1:10) were prepared according to the procedures described above. Samples were inoculated and then incubated statically for 72 hr at room temperature. To measure the relative transcript levels of *vpsL*, *ctxA*, and *tcpA* at the air–liquid interface, 500 μL of liquid culture at the surface of the culture and any present pellicle were extracted at 24, 48, and 72 hr. Samples were treated with TRIzol LS (Invitrogen) and RNA was extracted using chloroform (Fisher Chemical), isopropanol (Fisher Chemical), and ethanol (Fisher Chemical). Extracted RNA was treated with TURBO DNase (Invitrogen) to remove DNA. Integrity and purity were assessed using NanoDrop (ND-1000, Thermo Fisher). cDNA synthesis was performed on 1 μg of RNA using High Capacity cDNA reverse transcriptase (Applied Biosystems) according to the manufacturer's protocol. Resulting cDNA was diluted 1:2, and RT-PCR was performed to measure CtxA, TcpA, and VpsL expression. The primers used were (CtxA-F) 5'-TTGGAGCATTCCCACAACCC-3', (CtxA-R) 5'-GCTCCAGC AGCAGATGGTTA-3'' – amplicon 109 bp (*Zhao et al., 2018*), (TcpA-F) 5'-CGCTGAGACCACACCC ATA-3', (TcpA-R) 5'-GAAGAAGTTTGTAAAAGAAGAACACG-3' – amplicon 103 bp (*Fykse et al., 2007*), (VpsL-R) 5'-CATTCGTCGAACATCGCTGG-3', (VpsL-F) 5'-GTAGCGATTCACTATGGTGCGA-3' – amplicon 130 bp, (groEL-F) 5'-ATGATGTTGCCCACGCTAGA-3', and (groEL-R) 5'-GGTTATCGCTGC GGTAGAAG-3' – amplicon 117 bp (*Fykse et al., 2007*). GroEL was used for housekeeping gene. Real-time PCR was performed using SYBR green (Invitrogen) with 0.3 μM of specific primer sets and 2 μL of cDNA sample. PCR amplification was conducted on the StepOne RT-PCR System (Applied Biosystems) with the following conditions: 95°C for 5 min, 40 cycles of 95°C for 5 s, 58°C for 10 s, and 72°C for 15 s, and a final melting temperature analysis of PCR products. Each qRT-PCR run included a no-template and water negative control. Each sample was performed in duplicate. The ΔΔCT method was used to calculate fold change in expression levels (*Livak and Schmittgen, 2001*).

## Statistics

All statistical analyses were performed on GraphPad Prism v9.2 (GraphPad Software, San Diego, CA). Error bars in the figures depict the median with a 95% confidence interval as indicated. Based on the experimental design, either standard *t*-test or Mann–Whitney test was used to compare treatment groups as indicated in each figure legend.

## Acknowledgements

We thank members of the Ng and Weil Labs for helpful discussions. We acknowledge Dr. Andrew Bridges for sharing the *vpsL* reporter strain, Dr. Aretha Fiebig for suggestions on imaging pellicles, and Dr. Ed Ryan for his assistance in reviewing the manuscript. AAW and W-LN received support from a Rozan Award from Tufts University School of Medicine for this project. AAW was supported by AI123494 and W-LN was supported by AI121337 from the National Institute of Allergy and Infectious Diseases (NIAID). JY was supported by DP2GM146253 from the National Institute of General Medical Sciences (NIGMS). JY holds a Career Award at the Scientific Interface from the Burroughs Wellcome Fund (1015763.02). ARS was supported by the Tufts Post-Baccalaureate Research Education (PREP) Program (R25 GM066567).

# Additional information

## Funding

| Funder | Grant reference number | Author |
|---|---|---|
| National Institutes of Health | AI121337 | Wai-Leung Ng |
| National Institutes of Health | AI123494 | Ana A Weil |
| National Institutes of Health | DP2GM146253 | Jing Yan |
| National Institutes of Health | R25 GM066567 | Abigail Rivera Seda |
| Burroughs Wellcome Fund | 1015763.02 | Jing Yan |

The funders had no role in study design, data collection and interpretation, or the decision to submit the work for publication.

## Author contributions

Kelsey Barrasso, Conceptualization, Formal analysis, Investigation, Methodology, Validation, Writing – original draft, Writing – review and editing; Denise Chac, Formal analysis, Investigation, Methodology, Validation, Writing – review and editing; Meti D Debela, Anjali Steenhaut, Abigail Rivera Seda, Chelsea N Dunmire, Investigation; Catherine Geigel, Conceptualization, Formal analysis, Investigation, Writing – review and editing; Jason B Harris, Firdausi Qadri, Project administration, Resources; Regina C Larocque, Resources; Firas S Midani, Data curation, Formal analysis, Software; Jing Yan, Conceptualization, Formal analysis, Investigation, Methodology, Resources, Supervision, Writing – original draft, Writing – review and editing; Ana A Weil, Wai-Leung Ng, Conceptualization, Formal analysis, Funding acquisition, Investigation, Methodology, Project administration, Supervision, Writing – original draft, Writing – review and editing

## Author ORCIDs

Firas S Midani [ID] http://orcid.org/0000-0002-2473-7758
Ana A Weil [ID] http://orcid.org/0000-0002-6170-4306
Wai-Leung Ng [ID] http://orcid.org/0000-0002-8966-6604

## Ethics

The previously published study from which Figure 1 is derived from ref (7) received approval from the Ethical Review Committee at the icddr,b and the institutional review boards of Massachusetts General Hospital and the University of Washington. Participants or their guardians provided written informed consent.

All animal experiments were performed at and in accordance with the rules of the Tufts Comparative Medicine Services (CMS), following the guidelines of the American Veterinary Medical Association (AVMA) as well as the Guide for the Care and Use of Laboratory Animals of the National Institutes of Health. All procedures were performed with approval of the Tufts University CMS (Protocol# B 2018-99). Euthanasia was performed in accordance with guidelines provided by the AVMA and was approved by the Tufts CMS.

## Decision letter and Author response

Decision letter https://doi.org/10.7554/eLife.73010.sa1
Author response https://doi.org/10.7554/eLife.73010.sa2

# Additional files

## Supplementary files

• Supplementary file 1. Raw and normalised counts of *Pa* and *Vc*. (a) Operational taxonomic unit raw and normalized abundance of *Vibrio cholerae* (*Vc*) in *Paracoccus aminovorans* (*Pa*) colonized and *Pa* noncolonized study participants with *Vc* infection. (b) Raw and normalized abundance of total, *Pa*,

and *Vc* operational taxonomic units in all study participants.

• Transparent reporting form

## Data availability

All data generated or analysed during this study are included in the manuscript and supporting file; Source Data files have been provided for Figures 1-3, 5, 7.

The following previously published dataset was used:

| Author(s) | Year | Dataset title | Dataset URL | Database and Identifier |
|---|---|---|---|---|
| Midani FS, Weil AA, Chowdhury F, Begum YA, Khan AI, Debela MD, Durand HK, Reese AT, Nimmagadda SN, Silverman JD, Ellis CN, Ryan ET, Calderwood SB, Harris JB, Qadri F, David LA, LaRocque RC | 2018 | Human Gut Microbiota Predicts Susceptibility to Vibrio cholerae Infection | https://www.ebi.ac.uk/ena/browser/view/PRJEB17860 | European Nucleotide Archive accession number, PRJEB17860 |

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
