## [Editor Report]

In this work, the authors study the previously reported positive association between the presence of the gut bacterium *Paracoccus aminovorans* and *Vibrio cholerae* during infection. They describe and image dual-species biofilm formed in vitro as well as enhanced *V. cholerae* gut colonization in the presence of *P. aminovorans* in a neonatal mouse model. Collectively, the authors conclude that *P. aminovorans* enhances biofilm formation by *V. cholerae*, which could explain the increased abundance of*P. aminovorans* in persons infected with *V. cholerae* in endemic areas, and may have impacted the course of infections in humans.

---

## [Decision Letter]

**Decision letter after peer review:**

Thank you for submitting your article "Impact of a human gut microbe on *Vibrio cholerae* host colonization through biofilm enhancement" for consideration by *eLife*. Your article has been reviewed by 3 peer reviewers, and the evaluation has been overseen by a Reviewing Editor and Wendy Garrett as the Senior Editor. The reviewers have opted to remain anonymous.

Based on this consultation, it was concluded that additional data are required to support the claim of the study, namely that *Paracoccus aminovorans* enhances biofilm production by *Vibrio cholerae*. In this context, the experts are keen to see transcriptional and/or protein-level data shat show changes in expression of the vps genes or those genes that encode for the accessory proteins (RbmA, RbmC, Bap1 etc). Furthermore, transcriptional or protein-level data of canonical virulence factors-encoding genes (ctx, tcp)/ proteins (CTX by ELISA or TcpA by WB for instance) should be provided, as those data would support the notion on increased virulence of Vc in the presence of Pa.

Essential revisions:

1) Provide data to show that Pa increases biofilm production through transcriptional or protein-level data. Indeed, as Midani et al. (2018) showed that conditioned medium from *P. aminovorans* significantly increased growth of *V. cholerae*, it seems essential for your study to distinguish between increased growth (and hence more biofilm in the pellicles) versus biofilm enhancement, as claimed in your manuscript.

2) Please evaluate the impact of Pa on the production of *V. cholerae*'s main virulence factors.

3) Please provide a reanalysis of the data shown in Figure 1 to show the abundance/normalized abundance of P.a. and V.c.

*Reviewer #1 (Recommendations for the authors):*

– In Figure 1A, the authors could determine the relative abundance of *V. cholerae* in the 22 infected individuals to demonstrate that there a positive correlation between the abundance of *P. aminovorans* and the abundance of *V. cholerae* in the 6 individuals with a significantly higher proportion of *P. aminovorans*. The authors could also determine the "Normalized Abundance" of *P. aminovorans* and *V. cholerae* in these samples.

– The authors could also determine CFU of *P. aminovorans* in their suckling mouse model of infection to determine whether there is a correlation between the abundance of *P. aminovorans* and the abundance of *V. cholerae* during infection of these mice. This would also allow them to determine whether the ratio of abundance in mice is also 1:1 as observed in human samples.

– In Supplemental Figure 1, the authors could analyze their microbiota sequencing data to determine whether members of the Proteobacteria Phylum are being displaced by *P. aminovorans* at the Family level.

– The authors could measure the expression of a *V. cholerae* virulence factor (such as tcpA) by qPCR to determine whether there is an increase in virulence factor expression in *V. cholerae* in the presence of *P. aminovorans* (Figure 2B and Figure 2C).

– In Lines 166-168, the authors state, "Our findings also illustrate that our approach to microbiome studies in humans (6, 7) can be used as a predictive tool to identify gut microbes that alter Vc virulence." The authors should replace the word "virulence" with "colonization" which seems more appropriate for the data shown.

*Reviewer #2 (Recommendations for the authors):*

1. Based on the results shown in Figure xx the authors are proposing that *P. aminovorans* increases the ability of *V. cholerae* to produce VPL (Vibrio polysaccharides). I think this is an interesting point towards the mechanistic understanding by which *P. aminovorans* can enhances biofilm formation of this pathogens. I think it would be nice to further obtain support for this hypothesis using a transcription reporter fusion to a vpl-associated promoter to obtain further support for this proposal.

*Reviewer #3 (Recommendations for the authors):*

1) The introduction would be greatly improved by being specific about the model in which previous studies were done, where *V. cholerae* is found within the intestine in each model, and how the association between *V. cholerae* and the microbe of interest was established in the study (i.e. stool cultures vs intestinal tissue). When discussing concordant vs discordant observations, these details are very important (eg line 141).

2) Figure 2: Figure 2A: Here the authors show the results of a colonization assay in which Pa is inoculated into mice every 12 hours for 48 hours. Were the measurements taken at 24 and 48 hrs performed before or after the 24- and 48-hour inoculations, respectively. This should be clarified. If taken immediately after inoculation, this may not demonstrate colonization. Figure 2B: Here is Vc inoculated directly after the 48h inoculation of Pa? If so, have the authors tested whether the earlier inoculations are relevant or necessary?

3) Line 227-245 and Figure 6: Because WGA is not specific for VPS and Pa are not specifically labeled, conclusions from these studies must be tempered.

4) Line 356-259: The conclusion that Pa activates production of VPS would be greatly strengthened if transcriptional activation of the vps genes could be shown. If transcriptional activation is not observed, there is a concern that increased WGA staining in biofilms is not sufficient to support this claim.

5) Figure 2B: This strain of V*. cholerae* is well-known to undergo spontaneous mutations that result in QS-incompetence and increased biofilm production. Is it possible that Pa is promoting a change at the chromosomal level? If not, *V. cholerae* isolated from the Vc + Pa pellicle should behave like WT Vc in monoculture.

6) Figure 7: Based on the authors' model, intestinal titers of Pa should be higher during WT infection as compared with that during ΔvpsL infection. Intestinal titers of Pa increase in response to VPS production by Vc would help to support the in vitro results, which suggest that Vc VPS recruits Pa to biofilm aggregates.

Reviewing Editor comments:

1) The authors should provide complementation data for the vps / rbmA / rbmC+bap1 mutant. If this isn't possible, they should at least Sanger sequence the luxO gene of all their strains (WT and the mutants) to ensure all strains are QS-proficient and not derived from a QS-impaired circulating mutant of the "WT" C6706 isolate (which is known to produced enhanced biofilms).

2) Do the authors have colonization data beyond the 24h time point? If these data are not available, the topic should be at least discussed (e.g., whether colonization is expected to be enhanced beyond the 24h time point).

3) The authors wrote "Unlike previous studies (9, 10), pre-treatment with antibiotics did not change the outcome of Pa colonization (not shown)." => this is an important and somewhat surprising finding for which the data should be provided.

---

## [Author Response]

Essential revisions:1) Provide data to show that Pa increases biofilm production through transcriptional or protein-level data. Indeed, as Midani et al. (2018) showed that conditioned medium from P. aminovorans significantly increased growth of *V. cholerae*, it seems essential for your study to distinguish between increased growth (and hence more biofilm in the pellicles) versus biofilm enhancement, as claimed in your manuscript.

We have added Figure 5C which shows *vpsL* expression results from *Vc* grown in coculture with *Pa* and *Vc* alone. qRT-PCR analyses demonstrated that the relative transcript levels of *vpsL* (the first and a representative gene in the *vps*II operon) are higher in *Vc* cocultured with *Pa* than in *Vc* monoculture. In addition, in the new Figures 6C and D, we have used a P*_vpsL_-mNeonGreen* reporter to show the activation of the *vps*II operon in the co-culture pellicle, compared to the *Vc* monoculture pellicle.

We also want to clarify the results in the previous study (Midani 2018). In that study, *Vc* yield was higher in the conditioned medium (CM) harvested from *Pa* than *Vc* yield in CM harvested from *Vc*. However, the growth yield of *Vc* in the CM harvested from *Pa* was never as high as *Vc* in fresh medium. Indeed, *Vc* growth is higher in CM from *Pa* harvested at low cell-density than in CM from *Pa* harvested at high cell-density (our unpublished results), suggesting that the growth enhancement observed in the previous study is due to growth-limiting nutrients present in the CM harvested from *Pa* which were exhausted in CM harvested from *Vc*. This is the underlying reason that our prior results seem to contrast with our current findings. This was confirmed in our measurement of the planktonic growth yield of *Vc* monoculture which is comparable to *Vc* growth yield in the *Vc-Pa* coculture (Figure 3A). Importantly, biofilm formation is not enhanced when *Vc* is grown in CM harvested from *Pa*. We have added this discussion to the revised manuscript to clarify.

2) Please evaluate the impact of Pa on the production of *V. cholerae*'s main virulence factors.

We have added new Figure 7—figure supplement 1 showing the relative transcript levels of *ctxA* and *tcpA* in *Vc* monoculture and *Vc-Pa* co-culture.

We measured virulence gene expression in regular LB medium (identical to the conditions used in the biofilm coculture experiments). Here, we noted that under these growth conditions the relative transcript levels of these virulence genes were low (e.g., compared to the relative levels of *vpsL* shown in Figure 5C). These results suggest that these are not ideal conditions for virulence gene expression measurement, based on prior knowledge, it is known that virulence-inducing conditions typically requires specific growth conditions (AKI) and with a shaking phase. However, shaking in AKI medium prevents proper interaction between the two species. Nonetheless, we found that *ctxA* and *tcpA* transcript levels are slightly higher in coculture (although only the *ctxA* comparison is statistically significant). We have edited the manuscript to include this information.

3) Please provide a reanalysis of the data shown in Figure 1 to show the abundance/normalized abundance of P.a. and V.c.

We have clarified in the manuscript that Figure 1 shows normalized abundance data, and we have added new Supplementary file 1 (containing two excel tables of raw data), which include the raw and normalized abundance for *Pa*, and for *Vc* in *Vc*-infected persons.

Reviewer #1 (Recommendations for the authors):– In Figure 1A, the authors could determine the relative abundance of *V. cholerae* in the 22 infected individuals to demonstrate that there a positive correlation between the abundance of P. aminovorans and the abundance of *V. cholerae* in the 6 individuals with a significantly higher proportion of P. aminovorans. The authors could also determine the "Normalized Abundance" of P. aminovorans and *V. cholerae* in these samples.

We have clarified that the data shown in Figure 1 is normalized relative abundance, and we have shown the full raw and normalized data in the new Supplementary file 1. Please see above for detailed explanation of the results.

– The authors could also determine CFU of P. aminovorans in their suckling mouse model of infection to determine whether there is a correlation between the abundance of P. aminovorans and the abundance of *V. cholerae* during infection of these mice. This would also allow them to determine whether the ratio of abundance in mice is also 1:1 as observed in human samples.

We have determined the temporal dynamics of *Pa* in the small intestine of infant mouse (Figure 2—figure supplement 2). *Pa* can be detected 12 hours after the last *Pa* inoculation (right before *Vc* infection at 0 hr). The *Pa* counts decreased over the next 24 hours and the rate of decrease is independent of the presence of *Vc*. At the early onset of the infection when *Pa* was still detectable (6 hrs after *Vc* infection), the ratio between *Vc* and *Pa* is variable but close to 10:1. We have included these new data in the revised Figure 2—figure supplement 2. Although we did not observe the 1:1 ratio, which is likely due to the difference in intestinal physiology between human and mouse, we believe our model is still sufficient to show the effect of the presence of *Pa* inside the small intestine on *Vc* host colonization.

– In Supplemental Figure 1, the authors could analyze their microbiota sequencing data to determine whether members of the Proteobacteria Phylum are being displaced by P. aminovorans at the Family level.

We have added Panel C in Figure 2—figure supplement 1 ((original Supplemental Figure 1) which shows the order-level taxonomic impact of *Pa* colonization) within the Proteobacteria phylum. Differences in the two groups were not found to be significant with FDR-adjusted significance testing at multiple taxonomic levels.

– The authors could measure the expression of a *V. cholerae* virulence factor (such as tcpA) by qPCR to determine whether there is an increase in virulence factor expression in *V. cholerae* in the presence of P. aminovorans (Figure 2B and Figure 2C).

We have added this assay to our results as described above. These results are now shown in Figure 7—figure supplement 1. Please see above for detailed explanation.

– In Lines 166-168, the authors state, "Our findings also illustrate that our approach to microbiome studies in humans (6, 7) can be used as a predictive tool to identify gut microbes that alter Vc virulence." The authors should replace the word "virulence" with "colonization" which seems more appropriate for the data shown.

This correction has been made in the revised manuscript.

Reviewer #2 (Recommendations for the authors):1. Based on the results shown in Figure xx the authors are proposing that P. aminovorans increases the ability of V. cholerae to produce VPL (Vibrio polysaccharides). I think this is an interesting point towards the mechanistic understanding by which P. aminovorans can enhances biofilm formation of this pathogens. I think it would be nice to further obtain support for this hypothesis using a transcription reporter fusion to a vpl-associated promoter to obtain further support for this proposal.

In response to the reviewer’s request, we have obtained a strain harboring *P_vpsl_-mNeonGreen* and we have repeated the pellicle imaging in the updated Figure 6C-D. Interestingly, subpopulations of *Vc* cells have elevated *vpsL* level when co-cultured with *Pa*. These data are consistent with our new qRT-PCR results showing higher *vpsL* expression (Figure 5C).

Reviewer #3 (Recommendations for the authors):1) The introduction would be greatly improved by being specific about the model in which previous studies were done, where V. cholerae is found within the intestine in each model, and how the association between V. cholerae and the microbe of interest was established in the study (i.e. stool cultures vs intestinal tissue). When discussing concordant vs discordant observations, these details are very important (eg line 141).

We have revised the manuscript to include the recommended additional information on the models discussed in the introduction.

2) Figure 2: Figure 2A: Here the authors show the results of a colonization assay in which Pa is inoculated into mice every 12 hours for 48 hours. Were the measurements taken at 24 and 48 hrs performed before or after the 24- and 48-hour inoculations, respectively. This should be clarified. If taken immediately after inoculation, this may not demonstrate colonization. Figure 2B: Here is Vc inoculated directly after the 48h inoculation of Pa? If so, have the authors tested whether the earlier inoculations are relevant or necessary?

We apologize for the ambiguity. To clarify, the animals were inoculated with four doses of *Pa* at 0, 12, 24, and 36-hour time points. In some animals, after two doses of *Pa* inoculation, the *Pa* count in the small intestine was determined at the 24-hr (i.e., 12 hrs after the second *Pa* inoculation). The *Pa* count was also determined after four doses of *Pa* inoculation at 48-hr (i.e., 12 hrs after the last *Pa* inoculation). *Vc* infection was performed 12 hrs after the last *Pa* inoculation, and we have shown that *Pa* was detectable in the small intestine at that time point prior to *Vc* infection. We have revised the text and figure legends to clarify.

3) Line 227-245 and Figure 6: Because WGA is not specific for VPS and Pa are not specifically labeled, conclusions from these studies must be tempered.

In response to the reviewer’s question, we have added negative controls (new Figure 4—figure supplement 1) showing that *Pa* cells alone or *Vc* Δ*vpsL* mutant does not show WGA staining under the current conditions. While it is true that WGA stains GlcNAC residues, for Gram-negative bacterial cells including *Vc* and *Pa*, the WGA lectin molecules do not pass through the outer membrane. Only when the outer membrane is significantly impaired will one see strong WGA signal. Indeed, we always observe dead cells with strong WGA signal but those are easily distinguished from the VPS signal.

Regarding the labeling of *Pa* cells, currently we do not have good genetic tools to express fluorescent proteins in *Pa*; therefore, we relied on the universal membrane labeling of FM 4-64 to detect their presence in the co-culture biofilm. Importantly, *Pa* cells also have a characteristic cocci shape that is distinct from *Vc*’s curved-rod shape. In the updated Figure 6A-B, we have provided clearer zoom-in image, in which *Pa* cells can be clearly distinguished from *Vc* cells by both the absence of SCFP3A signal and the characteristic cocci shape.

4) Line 356-259: The conclusion that Pa activates production of VPS would be greatly strengthened if transcriptional activation of the vps genes could be shown. If transcriptional activation is not observed, there is a concern that increased WGA staining in biofilms is not sufficient to support this claim.

We have added new Figure 5C showing qRT-PCR results and Figure 6C-D showing microscopy analysis on *vpsL* expression from *Vc* grown in monoculture or in coculture with *Pa*. Please see above for detailed explanation.

5) Figure 2B: This strain of V. cholerae is well-known to undergo spontaneous mutations that result in QS-incompetence and increased biofilm production. Is it possible that Pa is promoting a change at the chromosomal level? If not, V. cholerae isolated from the Vc + Pa pellicle should behave like WT Vc in monoculture.

Thank you for pointing out this important issue. We have isolated 15 individual *Vc* colonies from pellicles harvested in *Vc* monocultures (5 isolates) and *Vc-Pa* cocultures (10 isolates). The *luxO* locus of these isolates was sequenced; all carry the WT *luxO* allele, suggesting *Pa* does not promote additional mutational change in *Vc* in this allele.

6) Figure 7: Based on the authors' model, intestinal titers of Pa should be higher during WT infection as compared with that during ΔvpsL infection. Intestinal titers of Pa increase in response to VPS production by Vc would help to support the in vitro results, which suggest that Vc VPS recruits Pa to biofilm aggregates.

We thank the reviewer for the great suggestion.

As discussed in the response to reviewer #1, we have determined the temporal dynamics of *Pa* in the small intestine of infant mouse. *Pa* can be detected 12 hours after the last *Pa* inoculation (right before *Vc* infection at 0 hr). The *Pa* counts decreased over the next 24 hours and the rate of decrease is independent of the presence of *Vc*. At the early onset of the infection when *Pa* was still detectable (6 hrs after *Vc* infection), the ratio between *Vc* and *Pa* is variable but close to 10:1. We have included these new data in the revised Figure 2—figure supplement 2. Although we did not observe the 1:1 ratio, which is likely due to the difference in intestinal physiology between human and mouse, we believe our model is still sufficient to show the effect of the presence of *Pa* inside the small intestine on *Vc* host colonization. Further studies are needed to fully understand the dynamic of the interaction in vivo.

Reviewing Editor comments:1) The authors should provide complementation data for the vps / rbmA / rbmC+bap1 mutant. If this isn't possible, they should at least Sanger sequence the luxO gene of all their strains (WT and the mutants) to ensure all strains are QS-proficient and not derived from a QS-impaired circulating mutant of the "WT" C6706 isolate (which is known to produced enhanced biofilms).

We have now provided new complementation data for the *vpsL* mutant in CV assay and pellicle assay (revised Figure 3—figure supplement 1), and in vivo (new Figure 7—figure supplement 2). All the other strains have been sequenced and they carry the WT allele of *luxO*. We also tested all our strains with a P*_qrr4_*-*lux* assay and they all show a Qrr4 expression pattern identical to the WT, confirming they carry a WT *luxO* allele.

2) Do the authors have colonization data beyond the 24h time point? If these data are not available, the topic should be at least discussed (e.g., whether colonization is expected to be enhanced beyond the 24h time point).

Our current IACUC protocol only allows us to keep the infant animals away from their dams for no more than 24 hours. Therefore, we could not obtain data points beyond 24 hours after infection. As predicted from our time course study (revised Figure 2—figure supplement 2), the difference between the *Vc* counts in mice with and without *Pa* steadily increases, therefore, we predict the difference could be bigger if we allow the *Vc* infection to continue beyond 24 hours.

3) The authors wrote "Unlike previous studies (9, 10), pre-treatment with antibiotics did not change the outcome of Pa colonization (not shown)." => this is an important and somewhat surprising finding for which the data should be provided.

Thank you for pointing out this ambiguity. We intended to point out that unlike previous studies in which antibiotic treatment was required to achieve bacterial colonization, in our study antibiotic treatment was not required for *Pa* colonization. We have shown *Pa* can colonize in the small intestines of untreated mice (Figure 2A). We have rewritten this statement to clarify.